# CtD: Composition through Decomposition in Emergent Communication

**Boaz Carmeli**
Technion – Israel Institute of Technology
`boaz.carmeli@campus.technion.ac.il,`

**Ron Meir**
Technion – Israel Institute of Technology
`rmeir@ee.technion.ac.il`

**Yonatan Belinkov**
Technion – Israel Institute of Technology
`belinkov@technion.ac.il`

## Abstract

Compositionality is a cognitive mechanism that allows humans to systematically combine known concepts in novel ways. This study demonstrates how artificial neural agents acquire and utilize compositional generalization to describe previously unseen images. Our method, termed "Composition through Decomposition", involves two sequential training steps. In the 'Decompose' step, the agents learn to decompose an image into basic concepts using a codebook acquired during interaction in a multi-target coordination game. Subsequently, in the 'Compose' step, the agents employ this codebook to describe novel images by composing basic concepts into complex phrases. Remarkably, we observe cases where generalization in the 'Compose' step is achieved zero-shot, without the need for additional training.

## 1 Introduction

A variety of communication methods can be found in nature (Munz, 2005; Meulenbroek & Morneault, 2022; Suzuki, 2016; Hölldobler, 1999). There are, however, many aspects of human language that are unique. One such aspect involves natural language's capacity to manage and process concepts. Humans are able to manipulate and compose concepts in a variety of ways. A chief ability among these is compositionality: the principle whereby new meanings arise from systematically combining simpler concepts in a novel manner (Frege, 1892; Li & Bowling, 2019; Hupkes et al., 2020). Through compositional generalization, humans can describe previously unseen objects and attribute meaning to unfamiliar scenes. However, neural networks struggle to achieve compositional representations (Lepori et al., 2023; Xu et al., 2022; Bouchacourt & Baroni, 2018; Dankers et al., 2022; Dankers & Titov, 2023). One reason for this disparity lies in neural networks' struggle to process and manipulate discrete entities. Several methods have been proposed and evaluated during recent years by the emergent communication (EC) research community to overcome this difficulty. Among them are reinforcement learning (RL) based optimization (Williams, 1992; Lazaridou et al., 2017), Gumbel-softmax (GS) sampling (Jang et al., 2017; Havrylov & Titov, 2017), and vector quantization (QT) (Carmeli et al., 2023). We describe these methods in more detail in Section 2

In this study, and consistent with previous work (Kottur et al., 2017), we emphasize that relying solely on discretization bias is insufficient for achieving compositionality. Additionally, we assert that before effectively composing basic concepts into complex ones, agents should acquire the ability to *decompose* complex concepts into basic ones. To demonstrate these claims, we train the agents to acquire a discrete concept codebook (Van Den Oord et al., 2017) through engagement in multi-target games (Mu & Goodman, 2021) (Figure 1, Left), which we expand upon in Section 2. Subsequently, we let them play a standard referential (Ref) game (Lazaridou & Baroni, 2020; Lazaridou et al., 2017) where they utilize the learned codebook to describe images using novel combinations of known 'words' (Figure 1, Right). We refer to our method as **"Composition through**

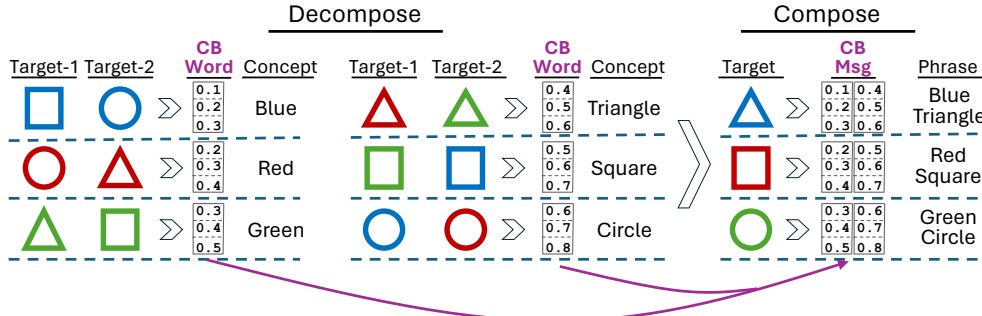

Figure 1: CtD training process: Concepts are learned and stored in the codebook during the DECOM-POSE step (Left), then composed to describe objects during the COMPOSE step (Right).

**Decomposion''** (CtD) and demonstrate its superior performance compared to existing methods by substantial margins. Notably, our method achieves perfect accuracy and compositionality, on multiple datasets. Learning a complex task by first learning a simpler task is reminiscent of the 'scaffolding' approach in developmental psychology (Wood et al., 1976) and of recent curriculum-based methods in machine learning (Soviany et al., 2022).

Our contribution comprises three essential elements for achieving compositional generalization: two that have proven useful in other setups and one introduced by us. **Multi-target:** We highlight the importance of multi-target coordination games in enabling neural agents to decompose complex objects into basic concepts. **Discrete codebook:** we establish the efficacy of employing a discrete codebook for representing these basic concepts in latent space. **CtD:** we introduce a novel two-step "composition through decomposition" approach, which leverages codebook-based communication in multi-target coordination games, surpassing current emergent communication methods in composition tasks and delivering perfect results on specific datasets.

## 2 EMERGENT COMMUNICATION AND COORDINATION GAMES

Much recent work studies the aspects of emergent communication (EC) by letting agents play a coordination game (Lewis, 1969) to accomplish a task that requires communication (Batali, 1998; Choi et al., 2018; Cao et al., 2018; Jaques et al., 2019; Das et al., 2019; Tian et al., 2020; Gratch et al., 2015; Yu et al., 2022). In the basic EC setup (Clark & Wilkes-Gibbs, 1986; Lazaridou et al., 2017; 2018), illustrated in Figure 2, there is a world $\mathbb{X}$, which is common to two agents: a sender $S$ and a receiver $R$. The sender and receiver observe $\mathbb{X}^S$ and $\mathbb{X}^R$, respectively. The world may be fully observable by both agents ($\mathbb{X} = \mathbb{X}^S = \mathbb{X}^R$) or partially observable ($\mathbb{X}^S, \mathbb{X}^R \subsetneq \mathbb{X}$).

Both agents have some representation of their observable world, $U^S$ and $U^R$. Given its current representation of the world, the sender $S$ sends a message $m$ over some communication channel to the receiver $R$. The message is a sequence of discrete elements over some vocabulary, which we call *words*. The receiver uses the message to identify the target(s) out of a set of candidate objects, for example identify a red triangle out of a set of objects with different shapes and colors.

Many game variants have been proposed in the EC literature (Lazaridou & Baroni, 2020; Lazaridou et al., 2017; Mu & Goodman, 2021; Dessì et al., 2021; Denamganaï & Walker, 2020). We provide a detailed description of the main ones in Appendix A.1. In this study, we focus on multi-target game variants, as outlined next.

### 2.1 MULTI-TARGET COORDINATION GAMES

In a multi-target coordination game (Mu & Goodman, 2021), the sender communicates about a *set of target objects* $\mathbb{T} = \{t_i\}$ out of a larger set of candidate objects $\mathbb{C}$. For instance, the target set may be all red triangles out of a larger candidate set of colorful shapes of varying sizes. Formally, assume a world $\mathbb{X}$, where each object is characterised by $n$ feature-value pairs (FVPs), $\langle f_1 : v_1, \ldots, f_n : v_n \rangle$, with feature $i$ having $k_i$ possible values, $v_i \in \{f_i[1], \ldots, f_i[k_i]\}$. For instance, the SHAPE world Kuhnle & Copestake (2017) has objects such as $\langle \text{shape:triangle,color:red} \rangle$.

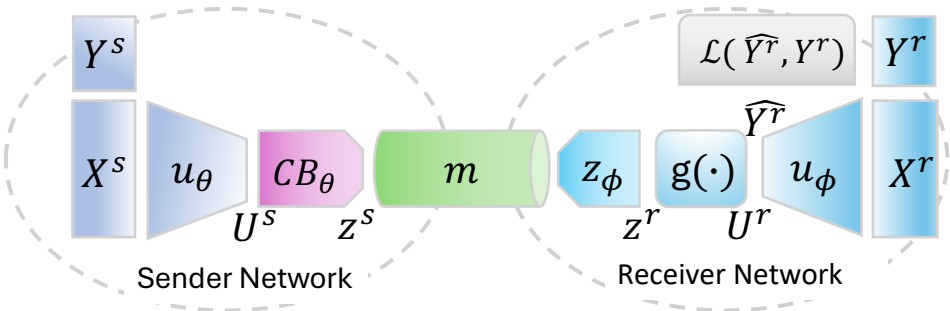

Figure 2: The emergent communicating architecture. Sender network is at the left, Receiver network is at the right, $m$ is the communication channel and $CB_\theta$ is the codebook.

Each "phrase" $l : \mathbb{X} \mapsto \{0, 1\}$ labels an object as $0$ or $1$. For instance, the phrase `Red Triangle` labels all red triangles as positive and all other objects as negative. We identify the phrase `Red Triangle` with the natural language (NL) phrase "red triangle", which comprises the basic *concepts* "red" and "triangle". At each turn of the game, a phrase is drawn at random from a set of labeling phrases $\mathbb{L}$, and a set of candidate objects $\mathbb{X}^s \subset \mathbb{X}$ is randomly selected, comprising both target objects, $\mathbb{T}$, adhering to the phrase, and distractors, $\mathbb{D}$ that do not: $l(x) = 1$ if $x \in \mathbb{T}$ and $l(x) = 0$ if $x \in \mathbb{D}$.

The sender encodes each of the targets $t_i \in \mathbb{T}$ into a continuous vector $\boldsymbol{u}_i^s \in \mathbb{R}^d$, and computes a target representative $\tilde{\boldsymbol{t}} = U^S = \frac{1}{n} \sum_{i=1}^{n} \boldsymbol{u}_i^S$. The receiver in a multi-target games is similar to that in single-target games. Refer to Appendix A.1 for details.

In this work we demonstrate that employing a multi-target setup is crucial for extracting concepts from images, calling into question the ability of compositionality to arise from traditional single-target games. Although multi-target games are seldom used in emergent communication studies, our experiments show that this setup is crucial for effective concept decomposition. Refer to Section 5.2 for the experimental results.

## 2.2 COMMUNICATION MODES

A key assumption in EC setups is that a discrete channel with restricted capacity serves as a facilitating bias for the emergence of compositionality (Havrylov & Titov, 2017; Lazaridou & Baroni, 2020; Vanneste et al., 2022). Thus, a large body of work has been concerned with enabling discrete communication in artificial multi-agent systems (Foerster et al., 2016; Lazaridou et al., 2017; Havrylov & Titov, 2017). However, the discretization requirement poses a significant challenge to multi-agent *neural* systems, which are typically trained with gradient-based optimization. Several approaches have been proposed for overcoming this challenge, e.g., relaxing the discrete communication with continuous approximations such as the Gumbel-softmax (GS) Jang et al. (2017); Havrylov & Titov (2017) or quantization (QT) Carmeli et al. (2023). More details on these methods are given in Appendix A.2.

In related studies, Tucker et al. (2022) suggest using a codebook to discretize the latent space, with a focus on balancing communication complexity and informativeness. However, their research does not explore compositional generalization, which is characterized by the ability to construct complex phrases from basic, meaningful concepts. Refer to Appendix H.4 for a detailed comparison.

In this work, we demonstrate how a communication protocol based on a latent discrete codebook (Van Den Oord et al., 2017), referred to as CB, can be used to develop a compositional language.

## 2.3 EVALUATING COMPOSITIONALITY

A large body of work has attempted to characterize EC in light of natural language traits, such as compositionality (Hupkes et al., 2020; Chaabouni et al., 2020), systematic generalization (Vani et al., 2021), pragmatism (Andreas & Klein, 2016; Zaslavsky et al., 2021), and more. In this work we are mainly interested is evaluating compositionality, defined by agents' ability to construct complex meanings from the structure and meaning of more basic parts (Li & Bowling, 2019; Andreas,

2019). A variety of metrics has been proposed in recent years Peters et al. (2024) for evaluating compositionality in emergent communication.

In this study we use five such evaluation metrics for compositionality: **Adjusted Mutual Information (AMI)** (Vinh et al., 2009) measures the mutual information between a set of EC messages and a corresponding set of NL phrases, adjusted for chance. **Positional Disentanglement (POS)** (Chaabouni et al., 2020) assesses how well words in specific positions within an EC message uniquely map to particular concepts in a NL phrase. **Bag of Symbols Disentanglement (BOS)** (Chaabouni et al., 2020) assesses the degree to which words in an EC language unambiguously correspond to NL concepts regardless of their position. **Context Independence (CI)** (Bogin et al., 2018) measures the alignment between EC words and NL concepts by examining their probabilistic associations across a set of EC messages and NL phrases. **Concept Best Matching (CBM)** (Carmeli et al., 2024) quantifies the best match between a set of EC messages and a corresponding set of NL phrases. We provide more information on these metrics in Appendix A.1.

## 3 COMPOSITION THROUGH DECOMPOSITION

In this section, we present the CtD approach. We begin by outlining the two-step training regimen of CtD, followed by its utilization of a codebook to decompose the latent space into a discrete set of vectors. Finally, we detail the multi-loss optimization process used to concurrently optimize the vector representations of the codebook and the task's success.

### 3.1 TWO-STEP TRAINING REGIME

The CtD approach is based on the idea of "Breaking down to build up". To achieve that, CtD involves two sequential training steps. First, in the DECOMPOSE step, agents are trained to learn a codebook with distinct sets of concepts using a multi-target SINGLE-CONCEPT dataset. (Refer to Section 2.1 for more details on multi-target games.) For example, to let the agent learn the concept `Blue`, we create a data sample with (at least) two targets: a `Blue Circle` and a `Blue Square`. See Figure 1 (Left). Second, in the COMPOSE step, we train the sender to describe an image from a COMPOSITE-PHRASE dataset by composing together a set of words from the learned codebook. For example, after learning the concepts `Blue` and `Triangle` during the DECOMPOSE step, the sender needs to compose a description `Blue Triangle` during the COMPOSE step. See Figure 1 (Right).

The two training steps differ in the data they use. Formally, each data sample $d_k \in D$ is a quadruple $\langle \mathbb{T}_k^s, \mathbb{D}_k^s, \mathbb{T}_k^r, \mathbb{D}_k^r \rangle$ composed from a set of the sender's targets and distractors and the receiver's targets and distractors, respectively. In the DECOMPOSE step, all targets share a phrase composed from exactly one concept (FVP). Unlike Mu & Goodman (2021), we do not use distractors in this step, demonstrating their redundancy for concept learning in a multi-target setup. In the COMPOSE step, all targets share a phrase composed from a set of concepts (FVPs) with a predefined length, $l$. Importantly, the sender's targets, $\mathbb{T}_k^s$, need not be the same as the receiver's $\mathbb{T}_k^r$, but all targets must share the same phrase. The number of targets and distractors may vary and is part of the configuration of each game.

Code-words are extracted from the codebook based on their similarity with the image representation $U^S$. During the DECOMPOSE step the top $l$ most similar vectors are extracted from the codebook, where $l$ is a dataset-dependent parameter. The sender then concatenates these $l$ vectors to form a bag-of-words message, $Z^s$.

### 3.2 LEARNING WITH A DISCRETE CODEBOOK

CtD uses a discrete codebook (Van Den Oord et al. (2017); Zheng & Vedaldi (2023)) to encourage the sender to transmit disentangled image features or concepts. The codebook consists of a set of learnable vectors (code-words), each representing a discrete concept in latent space, which we term *word*. Discretization is achieved by replacing an input representation vector with the closest word in the learned codebook. The words in the codebook are updated during training to best represent the distribution of data. A key advantage of using a codebook over other discretization methods is its ability to leverage a discrete latent space, learned during the DECOMPOSE step, for communicating

Table 1: Number of concepts and phrases in the experimental datasets for the two training phases.

| | SINGLE-CONCEPT | | COMPOSITE-PHRASE | | | | | |
|---|---|---|---|---|---|---|---|---|
| Dataset | Unique Concepts | Image Objects | Train Phrases | Val Phrases | Test Phrases | Image Objects | Phrase Length | Uniquely Identified |
| THING | 50 | 1 | 25024 | 950 | 958 | 1 | 5 | yes |
| SHAPE | 17 | 1 | 203 | 33 | 34 | 1 | 4 | no |
| MNIST | 10 | 1-2 | 70 | 15 | 15 | 2 | 2 | no |
| COCO | 80 | 1 | 261 | 14 | 26 | 2-3 | 2 | no |
| QRC | 50 | 1 | 25024 | 950 | 958 | 1 | 5 | yes |

compositional phrases during the COMPOSE step. We provide a formal definition of the codebook in Appendix E.

### 3.3 TRAINING WITH MULTIPLE-LOSS FUNCTION

We train the system end-to-end by minimizing the loss: $\mathcal{L} = \mathcal{L}_t + \beta_1 \mathcal{L}_c$. The task loss $\mathcal{L}_t$ is game-specific and typically involves either cross-entropy (CE) for MREF games, binary cross-entropy (BCE) for DIFF games, or mean squared error (MSE) for RECON games. Minimizing it leads to high accuracy on the game. The codebook loss $\mathcal{L}_c$ (Oord et al., 2018) is optimized by minimizing the MSE between the latent vectors $z$ and the code-words $w$,

$$\mathcal{L}_c = ||\text{sg}[z] - w||_2^2 + \beta_2 ||z - \text{sg}[w]||_2^2, \tag{1}$$

where $\text{sg}(\cdot)$ denotes the stop gradient function, needed to overcome the discreteness of the codebook and $\beta_2$ is a hyper-parameter that balances the two codebook loss components. Refer to Appendix G for more details.

The codebook loss encourages the sender's output representation, $U^S$ (see Fig. 2), to be close to the nearest word in the codebook. This is accomplished by utilizing the stop gradient function, which effectively employs a "straight-through" estimator. This mechanism allows gradients to flow through the quantization step during back-propagation (Van Den Oord et al., 2017). Minimizing the codebook loss often leads to high compositionality score as measured by the CBM and AMI metrics.

The two losses capture distinct aspects of the system and, as shown in Section 5, often compete with each other. We experimented with various values of $\beta_1$ and $\beta_2$, ranging from 0.1 to 1.0 and found no significant difference in results. To ensure consistency across all game setups, we set both to 1.0 for all experiments. Refer to Appendix G for details.

## 4 EXPERIMENTAL SETUP

Our experimental setup follows the two-step training regime outlined by the CtD method. Firstly, we train the agents to DECOMPOSE. For that we construct a SINGLE-CONCEPT dataset where all target share a single concept. Subsequently, we train the agents to COMPOSE using a COMPOSITE-PHRASE dataset where targets share a set of concepts. We compare the CtD approach with a C/D regime, in which agents are trained to COMPOSE without prior training them to DECOMPOSE. We also explore a variant in which only a single target is used during the COMPOSE step, akin to the classic REF game.

We evaluate game performance based on the accuracy of the agents in identifying the target(s) and the level of compositionality observed in their communication, as measured by the compositionality metrics.

### 4.1 DATA SPECIFICATIONS

**Datasets** We use five datasets in our experiments summarized in Table 1. THING (Carmeli et al., 2024) is a synthetic dataset of 100K objects encoded by concatenating five one-hot vectors. This dataset is best suited for evaluating compositionality in a highly controlled environment. SHAPE (Kuhnle & Copestake, 2017) is a visual reasoning dataset of colorful objects over a black background

.This dataset requires learning long phrases from images and is widely used by the EC community. MNIST (LeCun et al., 1998) is a dataset of handwritten digits, ranging from 0 to 9. For evaluating compositionality, we created a 2-digit version of the dataset by combining two images from the original dataset. Refer to Figure 9 for an example. COCO (Lin et al., 2014a) is a dataset of real-world multi-object images with categorical captions. QRC is a synthetic image dataset introduced by us, employing two-dimensional QR codes to encode information akin to the THING dataset. We use this dataset to evaluate results on a non-compositional dataset. We provide more information on each of these datasets in Appendix B.

**Rule construction**    Each data sample in the dataset comprises targets and distractors. These samples adhere to a labeling phrase, dictating the shared concepts among the targets. We generate two variants for every dataset, a SINGLE-CONCEPT and a COMPOSITE-PHRASE. In the SINGLE-CONCEPT, the labeling phrase consists of a single concept. The length of the labeling phrase in the COMPOSITE-PHRASE varies across datasets. Refer to Table 1 for details. Importantly, during training we set the message length to match the length of the phrases in the dataset. For example, we set the message length of the labeling phrase `Red Triangle` to 2. Constraining the message length to match the phrase length in the data restricts CtD's ability to learn variable-length communication. This limitation is further discussed in Appendix F and in Appendix J (Limitations).

**Data splits**    In the SINGLE-CONCEPT variant we split the data randomly into training, validation, and test sets. This approach aligns with our belief that agents cannot generalize across concepts; for example, learning the concepts `Red` and `Triangle` does not enable the agents to learn the concept `Square`. However, in the COMPOSITE-PHRASE variant, we ensure that there is no overlap between phrase labels in each split. Specifically, if a labeling phrase such as `Red:Square` appears in the test set, it does not appear in the training set. This segregation allows us to evaluate the extent to which agents have learned to generalize by composing known concepts in novel ways. In all our experiments we use 30,000 samples for training, 1000 for validation and 1000 for test.

## 4.2    COMMUNICATION BASELINES

We compare our results with two communication baselines, Gumbel-softmax (GS) (Havrylov & Titov, 2017; Jang et al., 2017) and Quantized (QT) (Carmeli et al., 2023). See Section 2 for details. We configured the GS vocabulary to match the number of concepts in each dataset. Additionally, for QT, we set the word length to enable the encoding of at least as many concepts as there are in the dataset. As expected, the two-step training regime did not yield any benefits for GS and QT, so we do not report those results.

## 4.3    TRAINING DETAILS

The EC setup encompasses numerous configuration and hyper-parameters that can be adjusted during experiments. To ensure a fair comparison, we modify only the parameters being tested. Specifically, we maintain consistency in the architecture of the agents ($u_\theta$ and $u_\phi$ in Figure 2) and network dimensions across all communication types. Furthermore, we select optimal vocabulary parameters for the different communication types. Agent architectures, $u_\theta$ and $u_\phi$, vary slightly across games to optimize the encoding of images into latent vectors $U^s$ and $U^r$. Refer to Appendix G for more details on the configuration and training parameters and procedures.

## 5    RESULTS

In this section, we present our experimental findings. We begin by outlining our primary results depicted in Figure 3. For detailed numerical results, please refer to Table 4 in Appendix C. In Section 5.2, we provide supplementary analyses and results from ablation experiments. Results for additional configurations and experiments are provided in Appendices D and E. We discuss the limitations of the CtD approach in Appendix J

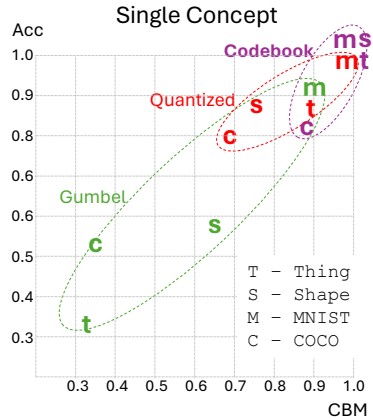 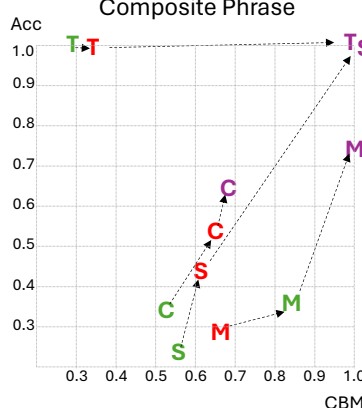

Figure 3: Accuracy and CBM results for the four compositional datasets (excluding the non-compositional QRC) and the three communication modes (Gumbel-softmax in green, Quantized in red and Codebook in purple) for the SINGLE-CONCEPT dataset (Left) and the COMPOSITE-PHRASE dataset (Right).

## 5.1 MAIN RESULTS

**Decomposition (D) performance**  Analyzing agents' performance on the SINGLE-CONCEPT dataset (Figure 3, Left) shows that codebook-based communication (indicated by purple letters in Figure 3 and the 'D' rows in Table 4 in Appendix C) achieves perfect accuracy (ACC) and compositionality (AMI, POS, BOS, CI and CBM) across the THING, SHAPE, and MNIST games. However, poorer results were observed for the COCO and non-compositional QRC datasets (Table 4 in Appendix C), which we further investigate below. In all setups, QT (represented by red letters in Figure 3) outperforms GS (green letters) in both accuracy and compositionality, as measured with CBM, while CB communication surpasses both.

**Composition through decomposition (CtD) performance**  Evaluating agents' performance on the COMPOSITE-PHRASE dataset (shown on the right side of Figure 3 and the 'CtD' rows in Tables 2 and 4) reveals that on the THING dataset, the CB-CtD approach achieves perfect accuracy, AMI, CI and CBM scores, and a near-perfect BOS score (0.98). On the SHAPE dataset CB-CtD achieves near-perfect accuracy (0.99), along with perfect CBM and CI. The BOS score for CB-CtD (0.84) significantly outperforms those of GS-CtD (0.06) and QT-CtD (0.11) by a wide margin. A high CI (0.929) and CBM (0.95) scores are also achieved for the MNIST dataset, showcasing agents' ability to utilize compositional communication. However, results are less conclusive for the COCO dataset (Table 4), which exhibits lower compositionality levels. Conversely, for the non-compositional QRC dataset, all compositionality metrics are notably low, and using an initialized codebook from the DECOMPOSE step degrades performance from 0.98 to 0.95 as shown in the bottom-right corner of Table 2. Notably, results for the GS and QT protocols, shown with green and red letters in Figure 3, are lower compared to CB (purple letters) across all compositional datasets illustrated in Figure 3, along both axes of this two-dimension space. Refer to Table 4 for detailed GS and QT results.

**Zero-shot (CtD-ZS) performance**  To underscore the agents' ability for compositional generalization, we train them on the SINGLE-CONCEPT dataset and evaluate their performance on the COMPOSITE-PHRASE dataset without further training. Comparing the 'CtD' and 'CtD-ZS' rows in Table 2 reveals that zero-shot (ZS) performance is comparable to, and in some cases (e.g., MNIST) surpasses, the results achieved with additional training. We attribute a portion of these striking results to the difficulty to further train the network on the COMPOSITE-PHRASE dataset with a multi-loss objective, as demonstrated in Figure 6. Notably, the zero-shot accuracy performance on the QRC dataset (0.48) is markedly lower compared to the CtD performance on this dataset (0.95). This unsurprising discrepancy indicates that learning a codebook during a dedicated DECOMPOSE step does not confer advantages for non-compositional datasets.

Table 2: Accuracy results and compositionality metrics for codebook-based communication and the COMPOSE step (excluding COCO). Depicting COMPOSE without DECOMPOSE (C/D), COMPOSE through DECOMPOSE (CtD) and zero-shot COMPOSE results after training the agents to DECOMPOSE (CtD-ZS). The results are averaged across five runs with different seeds. The highest standard deviations are 0.117, 0.035 and 0.050 for C/D, CtD and CtD-ZS, respectively.

| | **THING** | | | | | | **SHAPE** | | | | | |
| | ACC | AMI | POS | BOS | CI | CBM | ACC | AMI | POS | BOS | CI | CBM |
|---|---|---|---|---|---|---|---|---|---|---|---|---|
| CB-C/D | 0.25 | 0.16 | 0.01 | 0.05 | 0.03 | 0.25 | 0.26 | 0.52 | 0.05 | 0.09 | 0.13 | 0.47 |
| CB-CtD-ZS | 1.00 | 1.00 | 0.05 | 0.98 | 1.00 | 1.00 | 0.81 | 0.67 | 0.06 | 0.84 | 1.00 | 1.00 |
| CB-CtD | 1.00 | 1.00 | 0.05 | 0.98 | 1.00 | 1.00 | 0.99 | 0.78 | 0.07 | 0.84 | 1.00 | 1.00 |

| | **MNIST** | | | | | | **QRC** | | | | | |
| | ACC | AMI | POS | BOS | CI | CBM | ACC | AMI | POS | BOS | CI | CBM |
|---|---|---|---|---|---|---|---|---|---|---|---|---|
| CB-C/D | 0.41 | 0.86 | 0.08 | 0.29 | 0.38 | 0.58 | 0.98 | 0.94 | 0.00 | 0.01 | 0.01 | 0.17 |
| CB-CtD-ZS | 0.89 | 0.95 | 0.25 | 0.26 | 0.97 | 0.96 | 0.48 | 0.96 | 0.02 | 0.37 | 0.27 | 0.52 |
| CB-CtD | 0.81 | 0.92 | 0.09 | 0.26 | 0.93 | 0.95 | 0.95 | 0.96 | 0.00 | 0.01 | 0.01 | 0.17 |

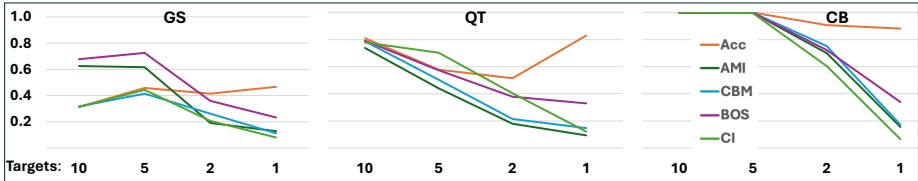

Figure 4: Accuracy, AMI, CBM, BOS and CI (Y-axis) versus number of targets (X-axis). Illustrating DECOMPOSE step for GS (Left) QT (middle) and CB (right) communication on the THING game.

**Composition without decomposition (C/D) performance** To further demonstrate the importance of the DECOMPOSE step for agents' ability to communicate complex objects during the COMPOSE step, we compare CtD accuracy and compositionality ('CB-CtD' rows in Table 2) against results achieved by the agents with an uninitialized codebook ('CB-C/D' rows in that table) across the COMPOSITE-PHRASE dataset. Notably, utilizing a codebook initialized during the DECOMPOSE step enables agents to achieve significantly improved results during the COMPOSE step. Specifically, with an appropriately initialized codebook agents achieve perfect accuracy, AMI, CI and CBM on the THING dataset, in contrast to accuracy, AMI, CI and CBM of 0.27, 0.172, 0.027 and 0.25, respectively, with an uninitialized codebook. Similar improvements were achieved for the SHAPE and MNIST datasets. Interestingly, on the non-compositional QRC dataset, accuracy reaches a near-perfect result of 0.98 even without an initialized codebook, and incorporating it did not improve the CBM score. In contrast, the GS and QT protocols did not show consistent performance improvements when using the two-step CtD approach compared to the single C/D step. Refer to Table 4 in Appendix C.2 for details.

## 5.2 AUXILIARY AND ABLATION RESULTS

In this subsection, we present results from several auxiliary and ablation experiments conducted to further clarify the outcomes obtained with the CB-CtD approach. Additional details and more ablation results are presented in Appendices C, D, E, and F.

**Learning with multiple targets** To assess the necessity of a multi-target setup we conducted an experiment where we varied the number of targets in the THING dataset during the DECOMPOSE step. As illustrated in Figure 4, reducing the number of targets enhances accuracy for the GS and QT protocols, and has a marginal negative impact when CB is used. However, there is a noticeable decline in all compositionality metrics, and transitioning from a two-target to a single-target setup for the CB protocol significantly reduces compositional performance, rendering it nearly random. Experiment results for the SHAPE and MNIST datasets are presented in Appendix D.2.

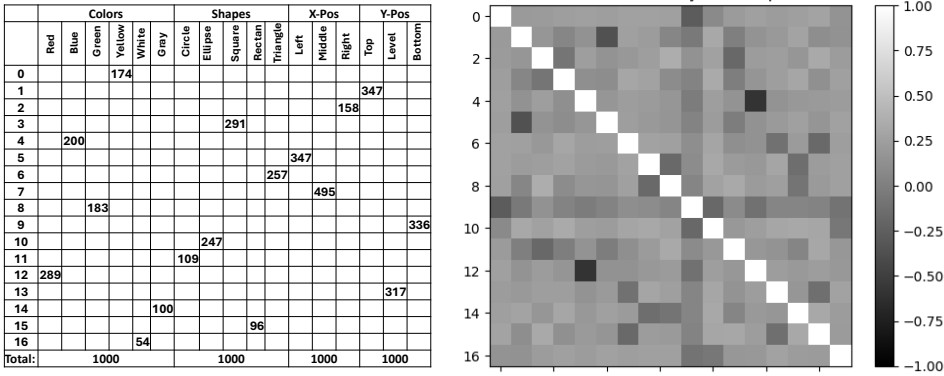

Figure 5: Left: Codebook utilization for the 17 SHAPE concepts trained on a COMPOSITE-PHRASE dataset and evaluated with CBM on 1,000 test samples, achieving a perfect score. Right: Heatmap of code-word similarities from the same run.

**Vocabulary size**  In this work, consistent with prior research (Mu & Goodman, 2021; Bogin et al., 2018), we set the vocabulary size, $|v|$, to match the number of basic concepts, $|c|$ present in the data. See Table 4 for detailed information on each dataset. The results show that the GS communication tends to use a subset of the vocabulary with an average ratio of $0.66$. The QT communication uses more words than there are concepts, enjoying a vocabulary to concept ratio of $1.21$. Conversely, the CB communication uses all words in the vocabulary, precisely matching the number of concepts in all but the COCO dataset. In Appendix E we show results for experiments involving an over-complete codebook that permits more words than concepts in the dataset. Future work may explore dynamic allocation algorithms to determine the optimal vocabulary size during training.

**Codebook utilization and visualization**  Figure 5 (Left) illustrates codebook utilization for the 17 SHAPE concepts trained on a COMPOSITE-PHRASE dataset and evaluated with CBM on 1,000 test samples. Each phrase in the test set is composed of 4 concepts, totaling $4,000$ concepts. As shown, each concept (top row) corresponds to exactly one codebook index (left column), resulting in a CBM score of $1.00$. Figure 5 (Right) presents a heatmap of code-word similarities, showing that the codes are generally well-separated, with the highest similarity score between any vector pairs being $0.35$. For additional details and an analysis of over-complete codebook utilization, refer to Appendix E.

**Multi-loss optimization**  Balancing between compositionality and task performance presents an optimization challenge. In Figure 6, we illustrate the training and validation losses (left Y-axis), and accuracy (right Y-axis) metrics for the THING (Top) and SHAPE (Bottom) games, comparing codebook initialization from scratch (Left) to initialization from a DECOMPOSE step (Right). In the case of the THING dataset (Top-left sub-figures for training and validation), commitment loss fails to converge when training from scratch, resulting in reasonable accuracy but poor compositionality. Initializing the codebook from the DECOMPOSE step (Top-right sub-figures) leads to perfect accuracy and minimal losses from the outset. The SHAPE experiment (Bottom-left sub-figures) demonstrates a different scenario. While both losses are minimized initially, overfitting occurs, indicated by the increasing task loss on the validation set (orange line, Bottom, second left sub-figure). However, initializing the system from a pre-trained codebook (Bottom-right sub-figures) prevents this phenomenon, resulting in near-perfect accuracy and compositionality.

# 6 RELATED WORK

**The necessity of multiple targets:**  Most studies exploring emergent compositional communication through referential games use a single-target setup. While many (Choi et al., 2018; Bogin et al., 2018) achieve high accuracy in extracting multi-word phrases from raw pixel datasets, none report a perfect compositionality score (see Appendix H.3 for more details). In this work, we show for the first time that both perfect accuracy and compositionality can be achieved for a multi-word image dataset. We argue that a multi-target setup is crucial for decomposing image representations into human-relevant

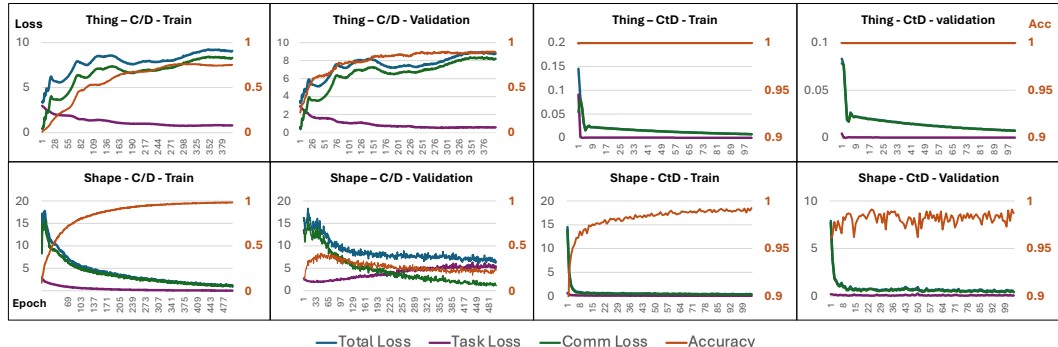

Figure 6: Multi-objective optimization. Top, THING game; Bottom SHAPE game. Left, training from scratch (CB); Right, starting from an initialized codebook (CtD). X-axis: number of epochs. We use different Y-axis values for optimal visualization.

concepts, though further research is needed to confirm this. Mu & Goodman (2021), the only known work using multiple targets, reports a maximal AMI score of 0.66, suggesting that while multi-target setup is likely necessary, it is not sufficient for compositionality to emerge.

**Codebook as a discrete communication channel** Vector quantization (Van Den Oord et al., 2017) and dictionary learning (Gregor & LeCun, 2010) are common methods to discretize and disentangle latent spaces (Hafner et al., 2020). Xu et al. (2022) found that increasing disentanglement pressure in $\beta$-VAE-based models (Higgins et al., 2017) degraded generalization for compositional tasks. Tucker et al. (2022) applied $\beta$-VAE in EC, showing that the information bottleneck principle achieves a human-like trade-off between communication complexity and task informativeness. However, they neither measured nor demonstrated compositionality. Refer to Appendix H.4 for a detailed comparison. In contrast, (Tamkin et al., 2023) proposed using VQ-VAE (Van Den Oord et al., 2017) for enforcing a discrete latent space in transformers (Vaswani, 2017), but did not explore its application in the context of EC. This technique revolves around online cluster learning (Zheng & Vedaldi, 2023), which, unlike $\beta$-VAE, introduces an architectural bias that encourages compositional generalization. Following Tamkin et al. (2023), we use VQ-VAE to discretize the EC communication channel and achieve fully compositional language.

**Iterated Learning** Iterated learning (IL) is a framework rooted in the concept of cultural transmission (Kirby, 2001; Kirby et al., 2008), modeling the transfer of knowledge across generations through a teacher-learner dynamic. In multi-agent neural IL, one agent generates data that another agent learns from, creating an iterative cycle where each learner becomes the teacher for the subsequent iteration. This chain-like structure mirrors the cultural evolution of language and knowledge, where compositionality is believed to play a pivotal role. Li & Bowling (2019) and Ren et al. (2020) were among the first to apply IL in emergent communication settings, using reinitialization of agent weights between iterations to simulate the iterative learning process.

## 7    CONCLUSIONS

In this work we demonstrate how neural networks can achieve compositionality similar to natural language, using a **dual-phase training approach** referred to as CtD. Through an initial DECOMPOSE phase, focused on concept learning, the network learns to meaningfully represent concepts in its latent space. Consequently, leveraging these learned concepts, it can accurately describe unseen complex objects even without further training. Our method assumes that there exists a foundational set of concepts upon which data is structured during the decomposition phase. An essential follow-up task is to demonstrate that such data can be self-represented by the sender without supervision. This capability raises an intriguing inquiry into identifying the most fundamental concepts that agents must grasp to facilitate the composition of novel meanings.

REPRODUCIBILITY

All datasets used in this work are either publicly available or can be easily generated using Python. A detailed description of each dataset is provided in Appendix B. Appendix A.3 offers a comprehensive explanation of the evaluation metrics, with Python implementations available online. Please refer to the respective papers for additional details. Appendix E details the codebook configuration and its hyperparameters. General training procedures and hyperparameter settings are outlined in Appendix G. All experiments were conducted on a single A100 GPU with 40 GB of RAM, and each experiment was completed in less than a day. The code for data preprocessing and all experiments will be made publicly available after the paper is accepted.

ACKNOWLEDGMENTS

This research was supported by grant no. 2022330 from the United States - Israel Binational Science Foundation (BSF), Jerusalem, Israel. BC and YB were supported by the Israel Science Foundation (grant no. 448/20), an Azrieli Foundation Early Career Faculty Fellowship, and an AI Alignment grant from Open Philanthropy. RM was partially supported by ISF grant 1693/22 and by the Skillman chair in biomedical sciences.

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

# Appendices

## A  COORDINATION GAME SETUPS

In this section we describe coordination game variants, communication protocols, and evaluation metrics in more detail.

### A.1  GAME VARIANTS

Many coordination game setups have been proposed and explored in recent years (Tian et al., 2020; Lazaridou et al., 2017; Leibo et al., 2017; Lazaridou et al., 2020; Lowe* et al., 2020; Choi et al., 2018; Qiu et al., 2022; Denamganaï & Walker, 2020). In this section, we elaborate on four game variants, summarized in Table 3. The referential (REF) game and the reconstruction (RECON) game represent the most common setups. DIFF is a multi-target game initially proposed by (Mu & Goodman, 2021) and MREF is our variant, which combines elements from both the REF and DIFF games.

**REF game**  This is the classical 'referential' game (Lazaridou et al., 2017). At each turn of the game, $n$ candidate objects $\mathbb{C} = \{c_i\}_{i=1}^n \subseteq \mathbb{X}$, are drawn uniformly at random from $\mathbb{X}$. One of them is randomly chosen to be the target $t$, while the rest, $\mathbb{D} = \mathbb{C} \setminus t$, serve as distractors. The sender $S$ encodes the target object $t$ via its encoder network $u_\theta$, such that $\boldsymbol{u}^s = u_\theta(t) \in \mathbb{R}^d$ is the encoded representation of $t$. It then uses its channel network $z_\theta$ to generate a message $m = z_\theta(u^s) = z_\theta(u_\theta(t))$. The channel and message have certain characteristics that influence both the emergent communication and the agents' performance in the game. The main communication modes are described in Appendix A.2.

At each turn, the receiver $R$ encodes each candidate object $c_i$ $(i = 1, 2, \ldots, n)$ via its encoder network $u_\phi$, to obtain $u_\phi(c_i)$. We write $U^r \in \mathbb{R}^{n \times d}$ to refer to the set of $n$ encoded candidate representations, each of dimension $d$. The receiver then decodes the message via its decoder channel network $z_\phi$, obtaining $\boldsymbol{z}^r = z_\phi(m) \in \mathbb{R}^d$. During training the entire system is optimized end-to-end with the cross-entropy loss based on similarity between the candidates $\mathbb{C}$ and the message $m$ such that:

$$\mathcal{L}_r^{CE} = -\log\left(\frac{\exp(z_\phi(m) \cdot u_\phi(x_i))}{\sum_{x \in \mathbb{C}} \exp(z_\phi(m) \cdot u_\phi(x))}\right) \tag{2}$$

where $x_i$ is the predicted target. Note that gradients from optimizing the contrastive loss $\mathcal{L}_r^{CE}$ allow the receiver to improve its latent world representation $U_\phi^r$ independently of the sender. At test time, the receiver computes a score matching each of the encoded candidates to the decoded message $\tilde{\boldsymbol{t}} = \mathrm{softmax}(z^r \cdot U^r)$, where 'softmax' is applied component-wise to $U^r = \{u_\phi(x_1), \ldots, u_\phi(x_n)\}$. The receiver's predicted target object is the one with the highest score, namely $\hat{t} = \mathrm{argmax}_i\{\tilde{t}_1, \ldots, \tilde{t}_n\}$.

**RECON game**  This is the well known Reconstruction game. In this variant, the receiver needs to reconstruct the target sent by the sender. At each turn the sender $S$ draws a target $t$ at random from $\mathbb{X}$, encodes it via its encoder network $u_\theta$, such that $\boldsymbol{u}^s = u_\theta(t) \in \mathbb{R}^d$, and uses its channel network $z_\theta$ to generate a message $m = z_\theta(u^s) = z_\theta(u_\theta(t))$. The receiver $R$ decodes the message $m$ via its decoder channel network $z_\phi$, obtaining $\boldsymbol{t}^r = z_\phi(m) \in \mathbb{R}^d$. During training the system is optimized

Table 3: Targets, distractors and loss function for game variants. 'Multiple' implies at least two.

| Game | Sender targets | Sender distractors | Receiver targets | Receiver distractors | Loss |
|------|------|------|------|------|------|
| REF | Single | No | Single | Yes | CE |
| RECON | Single | No | Single | No | MSE |
| DIFF | Multiple | Yes | Multiple | Yes | BCE |
| MREF | Multiple | No | Single | Yes | CE |

end-to-end with the mean square error (MSE) loss between $t^s$ and $t^r$ such that

$$\mathcal{L}_r^{\text{MSE}} = \frac{1}{n} \sum_{i=1}^{n} \|\boldsymbol{t}_i^s - \boldsymbol{t}_i^r\|^2. \tag{3}$$

At test time, $n-1$ distractor objects $\mathbb{D} = \{d_i\}_{i=1}^{n-1} \subseteq (\mathbb{X} \setminus t)$, are drawn from $\mathbb{X}$ and a candidate set $\mathbb{C}$ is constructed by $\mathbb{C} = \{c\}^n = \mathbb{D} \cup t$. The receiver computes a score matching each of the candidates $c_i$ to the decoded target $t^r$ using some distance metric $d(x_i, x_j)$ such as negative cosine similarity. Accuracy is then computed as in the REF game. This variant does not use distractors. It is based on the assumption that compositionality will emerge simply by posing a bottleneck on the communication channel. Prior work (Kottur et al., 2017; Bouchacourt & Baroni, 2018) show the limitation of this assumption.

**DIFF game**   DIFF is a **multi-target** game variant suggested by Mu & Goodman (2021). In this game variant the sender needs to communicate about a *set of target objects* $\mathbb{T} = \{t_i\}$ out of a larger set of candidate objects $\mathbb{C}$. For instance, the target set may be all red triangles out of a larger candidate set of colorful shapes. Formally, the DIFF game assumes a world $\mathbb{X}$, where each object is characterised by $n$ feature–value pairs (FVPs), $\langle f_1 : v_1, \ldots, f_n : v_n \rangle$, with feature $i$ having $k_i$ possible values, $v_i \in \{f_i[1], \ldots, f_i[k_i]\}$. For instance, the Shape world (Kuhnle & Copestake, 2017) has objects like $\langle$shape:triangle,color:red$\rangle$. Refer to Figure 8 for an example. Labeling phrases are boolean expressions over these FVPs. In this work we only use conjunctive expressions.

Each phrase $l : \mathbb{X} \mapsto \{0,1\}$ labels each object as 0 or 1. For instance, the phrase Red Triangle labels all red triangles as positive and all other objects as negative. We identify the phrase Red Triangle with the NL *phrase* "red triangle", which comprises the *concepts* "red" and "triangle". At each turn of the game, a phrase is drawn at random from a set of labeling phrases $\mathbb{L}$, and a set of candidate objects $\mathbb{X}^s \subset \mathbb{X}$ is randomly selected, comprising both target objects, $\mathbb{T}$, adhering to the phrase, and distractors, $\mathbb{D}$, which do not: $l(x) = 1$ if $x \in \mathbb{T}$ and $l(x) = 0$ if $x \in \mathbb{D}$.

The sender encodes each of the targets $t \in \mathbb{T}$ into a vector representation $\boldsymbol{u}^s \in \mathbb{R}^d$, and compute a target representative $\tilde{t} = \frac{1}{n} \sum_{i=1}^{n} t_i$. Similarly, a distractor representative $\tilde{d}$ is computed by averaging over the distractors. The sender then concatenates $\tilde{t}$ and $\tilde{d}$ and uses its channel network $z_\theta$ to generate a message $m = z_\theta(u^s) = z_\theta(u_\theta(\langle \tilde{t}, \tilde{d} \rangle))$, where $\langle \cdot, \cdot \rangle$ is the vector concatenation operator.

The receiver encodes each candidate object $c \in \mathbb{X}^r$ into a vector representation $\boldsymbol{u}_c^r = u_\phi(c) \in \mathbb{R}^d$. It decodes the message $m$ into a representation $\boldsymbol{z}^r = z_\phi(m) \in \mathbb{R}^d$ and computes a matching score between each encoded candidate and the message representation, $g(\boldsymbol{z}^r, \boldsymbol{u}_c^r)$. During training the system is optimized end-to-end with a binary cross entropy on correctly identifying each target:

$$\mathcal{L}_r^{\text{BCE}} = -\frac{1}{N} \sum_{i=1}^{|\mathbb{C}|} [y_i \cdot \log(\sigma(\hat{y}_i)) + (1 - y_i) \cdot \log(1 - \sigma(\hat{y}_i))] \tag{4}$$

where $y_i$ is the label of candidate $c_i$, $\hat{y}_i = g(\boldsymbol{z}^r, \boldsymbol{u}_{c_i}^r)$ and $\sigma(\hat{y}) = \frac{1}{1+e^{-\hat{y}}}$. A candidate is predicted as a target if its score $\sigma(\hat{y}) > 0.5$. Notably, as receiver observes multiple targets cross-entropy loss cannot be applied. The DIFF setup is more conducive to emergence of compositional communication, since the sender needs to form a generalization, rather than merely transmit the identity of a single object. Importantly, this variant requires the sender to communicate information about both the targets and distractors, allowing the receiver to discern whether a candidate aligns more closely with the target or distractor set.

**MREF game**   This is a **multi-target variant suggested by us** to minimize the required supervision and to align the DIFF game more closely with the classical REF game. In contrast to the DIFF variant, the sender in the MREF game does not use distractors. At each turn of the game, a phrase $l$ is drawn at random from a set of labeling phrases $\mathbb{L}$, and a set of target objects $x \in \mathbb{T} \subset \mathbb{X}$ where $l(x) = 1$ is drawn from $\mathbb{X}$. The sender calculates a target representative $\tilde{t}$ as in the DIFF game and communicates it to the receiver.

The MREF receiver is similar to the REF receiver. It observes a single target $t$ and a set of distractors $\mathbb{D}$ and uses cross-entropy loss to distinguishes between them. The MREF game differs from the REF game solely in its use of multiple targets by the sender. We contend that employing multiple targets during the DECOMPOSE step is crucial for compositional communication to emerge.

## A.2 COMMUNICATION MODES

In this section we describe several communication modes used in emergent communication setups. Each communication mode is characterized by its message encoding method and the channel capacity it permits.

**Continuous communication**    (CN) represents messages as floating point vectors. Though continuous, one may think of each vector element as if it represents a concept, and the vector itself represents a phrase. CN is expected to lead to good performance, provided that the channel has sufficient capacity. With CN, the system can easily be trained end-to-end with back-propagation. CN precludes the emergence of a discrete language, let alone compositional one. Thus, one may use it to assess an accuracy upper-bound without expecting a high compositionality score.

**Reinforcement learning**    (RL) addresses discreteness by employing RL algorithms (Williams, 1992; Lazaridou et al., 2017). RL-based communication protocols have demonstrated inferior performance in referential games compared to other methods (Havrylov & Titov, 2017) and are therefore not elaborated on further in this study.

**Gumbel-softmax**    (GS) is a continuous approximation for a categorical distribution. In the communication context, a discrete message is approximated via a sampling procedure. Details are given in Havrylov & Titov (2017); Jang et al. (2017). The GS end result is a multi-word message, where each word has the size of the alphabet and holds *one* (approximate) concept. GS allows for end-to-end optimization with gradient methods, and for discrete communication at inference time. However, the channel capacity is limited, and a large alphabet size is both inefficient (due to the need to sample from a large number of categories) and does not perform well in practice.

**Quantized-communication**    (QT) is discrete approximation to a continuous message (Carmeli et al., 2023). It uses a differentiable rounding approach to generate discrete vectors. In all our experiments we set the scaling factor to $1.0$, thus, effectively generate binary vectors. In contrast to GS, QT allows significant channel capacity even at limited word lengths.

## A.3 EVALUATION METRICS

In this subsection we provide more details on the five evaluation metrics used in this work.

**Adjusted Mutual Information (AMI)**    measures the mutual information between a set of EC messages and a corresponding set of NL phrases, adjusted for chance (Mu & Goodman, 2021). Given a test set $D$, let $\mathbb{M}$ denote the set of messages generated by the sender and $\mathbb{L}$ the set of NL phrases (e.g., `Red Triangle`, `Blue Square`) that appear as labels in the data. Formally, AMI (Vinh et al., 2009) is calculated by

$$\text{AMI}(M, L) = \frac{I(M, L) - \mathbb{E}(I(M, L)}{\max(H(M), H(L)) - \mathbb{E}(I(M, L))}, \tag{5}$$

where $I(M, L)$ is the mutual information between $M$ and $L$, $H(\cdot)$ is the entropy, and $\mathbb{E}(I(M, L))$ is computed with respect to a hypergeometric model of randomness. See (Vinh et al., 2009) for motivation for this distribution.

**Positional disentanglement (posdis)**    measures whether words in **specific positions** within EC messages consistently refer to a particular NL concept (Chaabouni et al., 2020).

Denoting by $w_j$ the $j$-th word of an EC message $m \in M$ and $c_1^j \in L$ the NL concept that has the highest mutual information, $I(\cdot\cdot)$, with $w_j$: $c_1^j = \arg\max_c I(w_j, c)$. In turn, $c_2^j$ is the second most informative attribute: $c_2^j = \arg\max_{c \neq c_1^j} I(w_j, c)$. Denoting $H(w_j)$ as the entropy of the $j$-th position, used for normalization, positional disentanglement (posdis) is defined as:

$$\text{posdis}(M, L) = \frac{1}{m_{len}} \sum_{j=1}^{m_{len}} \frac{I(w_j, c_1^j) - I(w_j, c_2^j)}{H(w_j)} \tag{6}$$

Where $m_{len}$ represents the length of the EC messages, which are all uniform in our case, and positions with zero entropy are excluded.

**Bag-of-symbols disentanglement (bosdis)** evaluates the extent to which words in the EC language unambiguously correspond to NL concepts, irrespective of their position within an EC message (Chaabouni et al., 2020).

Denoting by $n_w$ a counter of the word $w$ in a message $m \in M$, and $c_1^w \in L$ the NL concept that has the highest mutual information, $I(\cdot, \cdot)$, with $n_w$: $c_1^w = \arg\max_c I(n_w, c)$. In turn, $c_2^w$ is the second most informative attribute: $c_2^w = \arg\max_{c \neq c_1^w} I(n_w, c)$. Denoting $H(n_w)$ as the entropy of $n_w$ used for normalization, bag-of-symbols disentanglement (bosdis) is defined as:

$$\text{bosdis}(M, L) = \frac{1}{|W|} \sum_{w \in W} \frac{I(n_w, c_1^j) - I(n_w, c_2^j)}{H(n_w)} \tag{7}$$

**Context independence (CI)** seeks to quantify the alignment between words $w \in M$ and concepts $c \in L$ by examining their probabilistic associations (Bogin et al., 2018). Specifically, $P(w|c)$ represents the probability that a word $w$ is used when a concept $c$ is present, while $P(c|w)$ denotes the probability that a concept $c$ appears when a word $w$ is used. For each concept $c$, the word $w_c$ most frequently associated with it is identified by maximizing $P(c|w)$: $w_c := \arg\max_w P(c|w)$ The CI metric is then computed as the average product of these probabilities across all concepts:

$$\text{CI}(M, L) = \frac{1}{|C|} \sum_{c \in L} P(w_c|c) \cdot P(c|w_c) \tag{8}$$

The CI metric ranges from 0 to 1, with 1 indicating perfect alignment, meaning that each word retains its meaning consistently across different contexts and is thus used coherently.

**Concept Best Matching (CBM)** quantifies the best match between a set of EC messages and a corresponding set of NL phrases. Recently introduced by Carmeli et al. (2024) this metric aims to establish an explicit one-to-one mapping between EC words and NL concepts. Consequently, it is well-suited for evaluating the compositionality of datasets comprising images paired with compositional NL captions. CBM constructs a bipartite graph of EC words and NL concept and employs the Hungarian algorithm (Kuhn, 1955; Hopcroft & Karp, 1973) to find the best matching pairs between them.

## B  DATASETS

In this section we provide more details on the five datasets used in this work.

**THING (Carmeli et al., 2024)** A synthetic dataset of $100,000$ objects which we design for controlled experiments. Each object in the dataset is composed of five attributes, each with $10$ possible values. Overall, the THING datasets contains 50 concepts. Refer to Figure 7 (a) for the complete concept list. Each object in the dataset is uniquely identified by five FVPs. In contrast to the other datasets, objects in the THING game are encoded by concatenating five one-hot vectors, one per FVP. There are 50 unique values and 4 communication words (SOS, EOS, PAD, UNK) in the vocabulary. Thus, the vector dimension representing an object is $d = 54 \times 5 = 270$. Each such vector uniquely identifies an object in the dataset.

**SHAPE (Kuhnle & Copestake, 2017)** A visual reasoning dataset of objects over a black background. Refer to Figure 8 for few examples. Following Carmeli et al. (2024), we use a version of the dataset that contain four attributes: the shape attribute with five values; the color attribute with six values; horizontal position with three values; and vertical position with three values. Overall, the SHAPE dataset contains 17 concepts, and a maximal labeling phrase of 4 FVPs which sum to 270 unique 4-concept labels. The dataset includes various images sharing the same label. For instance, the two images on the left side of Figure 8 depict a `yellow square` positioned at the `left-center` side of each image.

| Shape | Color | Size | Age | Material |
|---|---|---|---|---|
| circle | red | petite | new | wood |
| ellipse | blue | tiny | vintage | metal |
| square | green | small | antique | plastic |
| rectangle | yellow | medium | modern | glass |
| triangle | white | large | classic | fabric |
| oval | gray | huge | retro | ceramic |
| pentagon | orange | miniature | contemporary | paper |
| hexagon | purple | gigantic | historic | leather |
| star | pink | massive | preowned | stone |
| heart | brown | enormous | timeless | rubber |

(a)

(b)

Figure 7: (a) The words used by the THING game and QRC games. Each object is composed from five words, one from each category. (b) QRC of 4 objects, each composed from 5 words. The top two QRCs in this illustrated example are the targets. They share the concept `triangle`. The bottom two QRCs serve as distractors, thus do not contain the `triangle` concept.

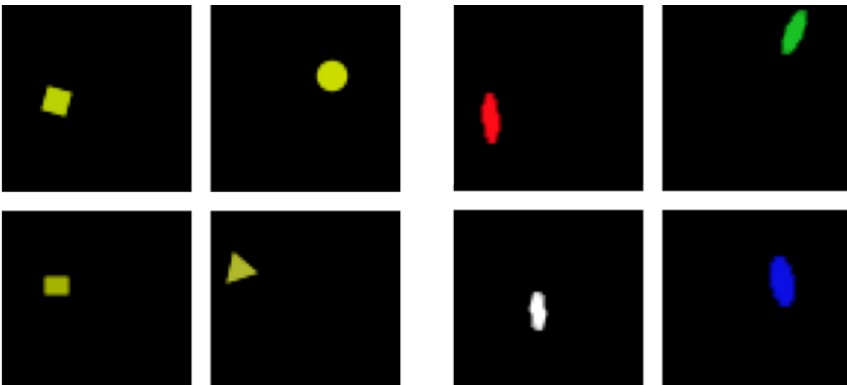

Figure 8: Two SHAPE examples from the SINGLE-CONCEPT dataset. (Left) `Yellow` concept. (Right) `Ellipse` concept.

**MNIST (LeCun et al., 1998)**  A widely-used dataset consisting of a collection of handwritten digits, ranging from 0 to 9, each captured as a gray-scale image with $28 \times 28$ pixels. For evaluating compositionality, we created a 2-digit version of the dataset by combining two images from the original dataset. See for example Figure 9, Right. To maintain similar image dimensions across the DECOMPOSE and COMPOSE steps, one of the images in each pair used for the DECOMPOSE step is a blank image with its location (right/left) randomly chosen. See Figure 9, (Left). Overall, the MNIST dataset contains 10 digits, and a maximal labeling phrase of 2 FVPs which sum to 100 unique 2-digit labels. The dataset contains many different images with identical labels.

**COCO (Lin et al., 2014b)**  Common Objects in Context is a widely-used benchmark dataset designed to capture real-world images containing multiple objects in various contexts. Each image is accompanied by a description of the objects within it. Overall the data contains 80 distinct object categories, namely concepts. Each image contains one or more identified objects. We filter the dataset to retain images with at most 3 objects. For the DECOMPOSE step we chose images that contain exactly one object. For the COMPOSE step we chose images with two or three distinct objects where we set the target images in each data sample to hold exactly two objects annotated with the same concepts. Refer to Figure 10 for few examples.

**QRC**  A synthetic dataset, introduced by us, which employs QR codes to encode information similar to the THING dataset. Each QRC in the dataset encodes five concepts, one from each category. Informationally, this dataset resembles the THING dataset. We experiment with it to evaluate whether

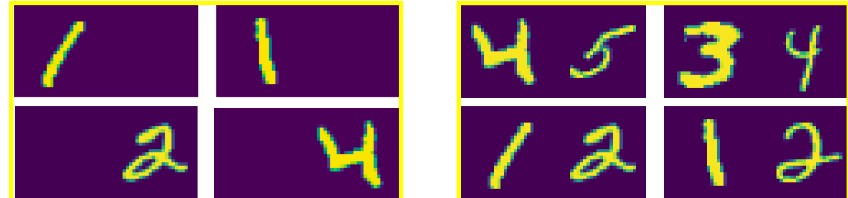

Figure 9: Training for decomposition using MNIST images. Concepts can be learned from images with a single digit (Left) or from images with two digits (Right). Illustration of two targets (Top) and two distractors (Bottom) for each data format. The target for the two-digit setting (Right) is '4'. Data from the two-digit setting is more similar to the data seen by the agents during the COMPOSE step. In the COMPOSE step, all targets in a data sample have the same two digits.

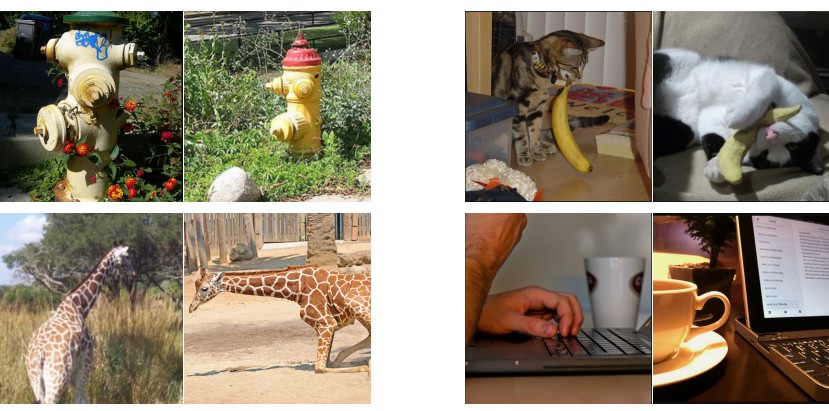

Figure 10: COCO dataset examples. (Left) SINGLE-CONCEPT. (Top-left) `fire hydrant`, (Bottom-left) `Giraffe`. (Right) COMPOSITE-PHRASE. (Top-right) `Cat` and `Banana`, (Bottom-right) `Cup` and `Laptop`.

encoding compositional information in a non-compositional manner, as is the case for QR codes, impedes the development of a compositional language by the agents. An example from the QRC dataset is depicted in Figure 7

## C    DETAILED RESULTS

In this section we provide detailed results for experiments described in the main body of the paper as well as for additional experiments we conducted to shade more light on various aspects of the CtD method compared to other approaches suggested in the literature.

### C.1    MREF GAME RESULTS

In Table 4 we provide detailed numerical results for the experiments shown in Figure 3. We analyse the results for the main experiments in Section 5. Below we describe complimentary results and additional insights.

We use the MREF setup for these experiments. In this setup the sender observes 20 targets and no distractors and the receiver observes one target and 20 distractors. Targets in each sample of the SINGLE-CONCEPT dataset share a single concept while targets at the COMPOSITE-PHRASE dataset share a composite phrase.

Rows in the table report results for experiments with five datasets (THING, SHAPE, MNIST, COCO, and QRC) and three communication protocols (GS, QT and CB) along three training regimes.For each dataset-protocol pair: The first row shows results for the DECOMPOSE phase ([comm]-D). The second row shows results for the COMPOSE phase, without initial decomposition training ([comm]-C/D). The third row shows results for the 'CtD' method ([comm]-CtD), where agents are trained on the

Table 4: Results for the MREF game. The sender observes 20 targets and no distractors, while the receiver observes one target and 20 distractors. Bold numbers indicates best results on the COMPOSITE-PHRASE dataset for each game.

| Dataset | comm | #v | #c | #p | l | #w | #m | ACC | AMI | POS | BOS | CI | CBM | #w/#c |
|---|---|---|---|---|---|---|---|---|---|---|---|---|---|---|
| THING | GS-D | 50 | 50 | 50 | 1 | 15 | 15 | 0.349 | 0.752 | 0.994 | 0.963 | 0.474 | 0.336 | 0.30 |
| | GS-C/D | 50 | 50 | 958 | 5 | 47 | 801 | 0.993 | 0.313 | 0.003 | 0.021 | 0.037 | 0.226 | 0.94 |
| | GS-CtD | 50 | 50 | 958 | 5 | 19 | 832 | 0.741 | 0.371 | 0.010 | 0.036 | 0.080 | 0.250 | 0.38 |
| | QT-D | 64 | 50 | 50 | 1 | 58 | 58 | 0.822 | 0.834 | 0.918 | 0.822 | 0.779 | 0.828 | 1.16 |
| | QT-C/D | 64 | 50 | 958 | 5 | 64 | 963 | 0.932 | 0.449 | 0.021 | 0.030 | 0.030 | 0.225 | 1.28 |
| | QT-CtD | 64 | 50 | 958 | 5 | 64 | 967 | 0.653 | 0.420 | **0.040** | 0.034 | 0.035 | 0.260 | 1.28 |
| | CB-D | 50 | 50 | 50 | 1 | 50 | 50 | 1.000 | 1.000 | 1.000 | 1.000 | 1.000 | 1.000 | 1.00 |
| | CB-C/D | 50 | 50 | 958 | 5 | 50 | 644 | 0.273 | 0.172 | 0.025 | 0.046 | 0.027 | 0.258 | 1.00 |
| | CB-CtD | 50 | 50 | 958 | 5 | 50 | 958 | **1.000** | **1.000** | 0.033 | **0.975** | **1.000** | **1.000** | 1.00 |
| SHAPE | GS-D | 17 | 17 | 17 | 1 | 14 | 14 | 0.569 | 0.612 | 0.678 | 0.576 | 0.551 | 0.570 | 0.82 |
| | GS-C/D | 17 | 17 | 34 | 4 | 16 | 187 | 0.230 | 0.579 | 0.028 | 0.053 | 0.135 | 0.548 | 0.94 |
| | GS-CtD | 17 | 17 | 34 | 4 | 17 | 104 | 0.135 | 0.717 | 0.011 | 0.066 | 0.240 | 0.609 | 1.00 |
| | QT-D | 32 | 17 | 17 | 1 | 23 | 23 | 0.813 | 0.881 | 0.881 | 0.692 | 0.762 | 0.770 | 1.35 |
| | QT-C/D | 32 | 17 | 34 | 4 | 31 | 322 | 0.413 | 0.568 | 0.063 | 0.063 | 0.125 | 0.560 | 1.82 |
| | QT-CtD | 32 | 17 | 34 | 4 | 27 | 123 | 0.100 | **0.757** | **0.227** | 0.110 | 0.205 | 0.714 | 1.59 |
| | CB-D | 17 | 17 | 17 | 1 | 17 | 17 | 1.000 | 1.000 | 1.000 | 1.000 | 1.000 | 1.000 | 1.00 |
| | CB-C/D | 17 | 17 | 34 | 4 | 17 | 396 | 0.213 | 0.532 | 0.041 | 0.103 | 0.139 | 0.47 | 1.00 |
| | CB-CtD | 17 | 17 | 34 | 4 | 17 | 192 | **0.988** | 0.745 | 0.045 | **0.841** | **0.999** | **1.00** | 1.00 |
| MNIST | GS-D | 10 | 10 | 10 | 1 | 9 | 9 | 0.886 | 0.966 | 1.000 | 1.000 | 0.947 | 0.898 | 0.90 |
| | GS-C/D | 10 | 10 | 15 | 2 | 10 | 26 | 0.360 | 0.773 | **0.242** | **0.288** | 0.433 | 0.806 | 1.00 |
| | GS-CtD | 10 | 10 | 15 | 2 | 10 | 23 | 0.799 | 0.904 | 0.086 | 0.280 | 0.779 | 0.862 | 1.00 |
| | QT-D | 16 | 10 | 10 | 1 | 11 | 11 | 0.999 | 0.999 | 0.998 | 0.933 | 0.998 | 0.999 | 1.10 |
| | QT-C/D | 16 | 10 | 15 | 2 | 15 | 45 | 0.294 | 0.719 | 0.159 | 0.136 | 0.324 | 0.638 | 1.50 |
| | QT-CtD | 16 | 10 | 15 | 2 | 12 | 23 | 0.391 | 0.736 | 0.173 | 0.211 | 0.344 | 0.768 | 1.20 |
| | CB-D | 10 | 10 | 10 | 1 | 10 | 10 | 0.999 | 1.000 | 1.000 | 1.000 | 1.000 | 1.000 | 1.00 |
| | CB-C/D | 10 | 10 | 15 | 2 | 10 | 36 | 0.422 | 0.837 | 0.047 | 0.287 | 0.377 | 0.539 | 1.00 |
| | CB-CtD | 10 | 10 | 15 | 2 | 10 | 32 | **0.811** | **0.919** | 0.086 | 0.260 | **0.929** | **0.950** | 1.00 |
| COCO | GS-D | 80 | 80 | 80 | 1 | 56 | 56 | 0.501 | 0.551 | 0.78 | 0.64 | 0.32 | 0.318 | 0.70 |
| | GS-C/D | 80 | 80 | 26 | 2 | 54 | 87 | 0.442 | 0.719 | 0.299 | 0.286 | 0.223 | 0.535 | 0.68 |
| | GS-CtD | 80 | 80 | 26 | 2 | 49 | 95 | 0.509 | 0.746 | 0.294 | 0.285 | 0.277 | 0.592 | 0.61 |
| | QT-D | 128 | 80 | 80 | 1 | 82 | 82 | 0.789 | 0.682 | 0.892 | 0.791 | 0.611 | 0.663 | 1.03 |
| | QT-C/D | 128 | 80 | 26 | 2 | 60 | 110 | 0.571 | 0.585 | 0.304 | 0.231 | 0.328 | 0.638 | 0.75 |
| | QT-CtD | 128 | 80 | 26 | 2 | 51 | 92 | 0.628 | 0.773 | **0.310** | 0.262 | 0.245 | **0.750** | 0.64 |
| | CB-D | 80 | 80 | 80 | 1 | 77 | 77 | 0.827 | 0.902 | 0.956 | 0.919 | 0.831 | 0.857 | 0.96 |
| | CB-C/D | 80 | 80 | 26 | 2 | 62 | 143 | 0.634 | 0.736 | 0.272 | 0.274 | 0.280 | 0.582 | 0.78 |
| | CB-CtD | 80 | 80 | 26 | 2 | 56 | 97 | **0.654** | **0.805** | 0.283 | **0.295** | **0.342** | 0.638 | 0.70 |
| QRC | GS-D | 50 | 50 | 50 | 1 | 1 | 1 | 0.069 | 0.000 | 0.000 | 0.000 | 0.028 | 0.028 | 0.02 |
| | GS-C/D | 50 | 50 | 958 | 5 | 31 | 214 | 0.766 | 0.041 | **0.007** | **0.011** | **0.026** | 0.173 | 0.62 |
| | GS-CtD | 50 | 50 | 958 | 5 | 5 | 1 | 0.203 | 0.000 | 0.000 | 0.000 | 0.005 | 0.116 | 0.00 |
| | QT-D | 64 | 50 | 50 | 1 | 50 | 50 | 0.528 | 0.685 | 0.828 | 0.727 | 0.572 | 0.594 | 1.00 |
| | QT-C/D | 64 | 50 | 958 | 5 | 64 | 968 | 0.973 | 0.514 | 0.002 | 0.010 | 0.013 | 0.170 | 1.28 |
| | QT-CtD | 64 | 50 | 958 | 5 | 62 | 555 | 0.814 | 0.077 | 0.002 | 0.010 | 0.014 | **0.222** | 1.24 |
| | CB-D | 50 | 50 | 50 | 1 | 50 | 50 | 0.485 | 0.756 | 0.839 | 0.723 | 0.479 | 0.643 | 1.00 |
| | CB-C/D | 50 | 50 | 958 | 5 | 50 | 954 | **0.985** | 0.955 | 0.003 | 0.007 | 0.011 | 0.168 | 1.00 |
| | CB-CtD | 50 | 50 | 958 | 5 | 50 | 958 | 0.956 | **1.000** | 0.001 | 0.006 | 0.013 | 0.172 | 1.00 |

COMPOSITE-PHRASE dataset using parameters initialized from the DECOMPOSE experiments in the first row.

The '#v' column reports the vocabulary size. For QT, vocabulary size must be to the power of 2. '#c' and '#p' indicate the number of unique NL concepts and phrases in the test set, respectively. The 'l' column reports phrase length. For the SINGLE-CONCEPT dataset $l = 1$. In the case of the COMPOSITE-PHRASE dataset, this parameter indicates the number of concepts used to describe a composite object in the dataset. '#w' and '#m' report the number of unique EC words and messages

generated by the sender, respectively. The 'ACC', 'AMI', 'POS', 'BOS', 'CI' and 'CBM', report the accuracy, adjusted-mutual-information, positional disentanglement, bag-of-symbols disentanglement, context independence and concept-best-matching, respectively. The #w/#c column reports the ratio between number of generated words (#w) to the number of concepts (#c) in the dataset. This ratio reveals interesting differences between communication protocols. Refer to Section 5.2 for details.

## C.2   GS AND QT CTD RESULTS

To further evaluate the importance of codebook-based communication in the CtD approach, we assess its performance using the GS and QT protocols. Specifically, we compare the performance of GS and QT when trained on the COMPOSITE-PHRASE dataset without prior initialization from the SINGLE-CONCEPT datasets (C/D) to their performance when the CtD method is applied.

For the THING dataset, the CtD accuracy for both GS (0.741) and QT (0.653) is lower than that of C/D (0.993 and 0.932, respectively), while the compositionality metrics remain roughly the same. Similar trends are observed for the SHAPE dataset. Results for the MNIST dataset are somewhat different. The CtD accuracy for both GS (0.799) and QT (0.391) is higher than that of C/D (0.36 and 0.294, respectively). We observed no substantial differences in compositionality metrics across the two protocols and three datasets.

We conclude that the CtD approach offers no advantage over C/D for the GS and QT protocols across any of the evaluated datasets.

## C.3   EVALUATING THE COMPOSITIONALITY METRICS

There are clear differences among the five metrics used to evaluate compositionality, despite all being designed to measure the same phenomenon.

The CBM metric is the only one not based on information theory; instead, it aims to maximize the matches between EC words and NL concepts. This makes it the most interpretable metric, with results that are easier to align with our expectations of the language that should emerge.

The CI metric shows a high correlation with CBM (0.89). The main distinction between the two is that CI is a statistical learning metric based on IBM Model 1 (Brown et al., 1993), utilizing the expectation-maximization algorithm (Dempster et al., 1977). Consequently, this metric requires training and is less interpretable, particularly when compositionality is low.

The AMI metric, used by Mu & Goodman (2021) to assess their multi-target study, also shows a high correlation with CBM (0.76) and CI (0.70). This metric relies on cluster labeling. The exceptionally high AMI scores for C/D (0.95) and CtD (1.0) on the non-compositional QRC dataset highlight a limitation of this metric that warrants further investigation.

The BOS metric, initially proposed by Chaabouni et al. (2020), relies on mutual information. It shows a stronger correlation with CI (0.86) compared to CBM (0.69). The metric assesses the difference between the most and second most informative concepts, which can potentially skew results, as evidenced by the CB-CtD results on the THING and SHAPE datasets.

The POS metric, which is used to assess positional disentanglement, yields low scores across all COMPOSITE-PHRASE datasets, with a maximum score of 0.342 observed for the COCO QT-CtD setup. Since we do not assume positional meaning in the messages, we find this metric less relevant to our experiments.

## C.4   DIFF GAME RESULTS

In Table 5 we report results for a series of experiments using the DIFF game setup, which is similar to the setup proposed by Mu & Goodman (2021). In contrast to the MREF game, agents in the DIFF game observe 20 targets and 20 distractors in each turn, and optimization is done with BCE loss. For further details, refer to Section A.1. The structure of Table 5, along with the description of its columns and rows, mirrors that of Table 4 and is detailed in Section C.1. The 'd' column, which is absent in the main table, denotes the dimension of the codebook's words and is independent of the dataset. Despite experimenting with various sizes, we did not observe significant differences among them.

Table 5: Results for the DIFF game with 20 targets and 20 distractors observed by the agents during both phases. The 'd' column indicates word dimension for each dataset.

| Dataset | comm | d | #c | #p | l | #w | #m | ACC | AMI | POS | BOS | CI | CBM | #w/#c |
|---------|------|---|----|----|---|----|----|-----|-----|-----|-----|----|-----|-------|
| THING | CB-D | 100 | 50 | 50 | 1 | 50 | 50 | 0.998 | 0.994 | 0.998 | 0.991 | 0.975 | 0.982 | 1.00 |
| | CB-C/D | 100 | 50 | 958 | 5 | 50 | 765 | 0.697 | 0.149 | 0.009 | 0.067 | 0.033 | 0.260 | 1.00 |
| | CB-CtD | 100 | 50 | 958 | 5 | 50 | 981 | **0.995** | **0.619** | **0.010** | **0.939** | **0.963** | **0.970** | 1.00 |
| SHAPE | CB-D | 200 | 17 | 17 | 1 | 17 | 17 | 1.000 | 1.000 | 1.000 | 1.000 | 1.000 | 1.000 | 1.00 |
| | CB-C/D | 200 | 17 | 34 | 4 | 17 | 591 | 0.670 | 0.339 | 0.021 | 0.067 | 0.107 | 0.445 | 1.00 |
| | CB-CtD | 200 | 17 | 34 | 4 | 17 | 297 | **0.890** | **0.633** | **0.061** | **0.649** | **0.761** | **0.909** | 1.00 |
| MNIST | CB-D | 100 | 10 | 10 | 1 | 10 | 10 | 0.996 | 1.000 | 1.000 | 1.000 | 1.000 | 1.000 | 1.00 |
| | CB-C/D | 100 | 10 | 15 | 2 | 10 | 58 | 0.771 | 0.677 | 0.159 | 0.217 | 0.250 | 0.555 | 1.00 |
| | CB-CtD | 100 | 10 | 15 | 2 | 10 | 23 | **0.964** | **0.920** | **0.251** | **0.282** | **0.925** | **0.947** | 1.00 |
| COCO | CB-D | 100 | 80 | 80 | 1 | 78 | 78 | 0.887 | 0.906 | 0.958 | 0.916 | 0.847 | 0.858 | 0.98 |
| | CB-C/D | 100 | 80 | 26 | 2 | 66 | 185 | 0.792 | 0.650 | 0.243 | 0.227 | 0.250 | 0.548 | 0.83 |
| | CB-CtD | 100 | 80 | 26 | 2 | 67 | 141 | **0.836** | **0.734** | **0.308** | **0.309** | **0.341** | **0.610** | 0.84 |
| QRC | CB-D | 400 | 50 | 50 | 1 | 50 | 50 | 0.929 | 0.954 | 0.975 | 0.940 | 0.894 | 0.894 | 1.00 |
| | CB-C/D | 400 | 50 | 958 | 5 | 50 | 986 | **0.916** | **0.272** | 0.001 | 0.007 | 0.011 | 0.166 | 1.00 |
| | CB-CtD | 400 | 50 | 958 | 5 | 49 | 936 | 0.899 | 0.194 | **0.004** | **0.012** | **0.016** | **0.167** | 0.98 |

While some differences in results exist between the two setups, qualitative observations remain similar. Specifically, with CtD, agents improve both accuracy and compositionality results across all datasets (except QRC) compared to a CB setup that does not utilize a learned codebook (C/D). CtD outperforms the GS and QT protocols in the THING, SHAPE, and MNIST datasets and achieves results comparable to QT on the COCO dataset. As observed previously, the non-compositional QRC dataset does not benefit from the CtD approach.

## C.5 DATA MISMATCH BETWEEN DECOMPOSE AND COMPOSE STEPS

The CtD training approach presents an optimization challenge due to potential data mismatch between the DECOMPOSE and COMPOSE steps. Figure 9 depicts two data options for the MNIST SINGLE-CONCEPT dataset. While training on a single digit (Figure 9, Left) may seem more intuitive, training with two digits in each image, where only one is the target (Figure 9, Right), is more similar to the data seen during the COMPOSE step. Indeed, training the DECOMPOSE step with the two-digit version resulted in better accuracy (0.810 vs. 0.710) and CBM (0.961 vs. 0.902) compared to the one-digit version, respectively. The mismatch between the SINGLE-CONCEPT and the COMPOSITE-PHRASE datasets is one of the reasons for the lower performance on the COCO dataset. Images for this game vary significantly between the two datasets.

## C.6 QRC: A NON-COMPOSITIONAL DATASET

The QRC dataset comprises synthetic QRC images encoding information akin to that of the THING dataset. Refer to Figure 7 for the complete list of THING concepts and a few QRC image examples. As such, this dataset presents an intriguing setting wherein images encode compositional information in a non-compositional way.

We utilize the QRC dataset to elucidate various aspects of the CtD approach. One such aspect, as reported in Bouchacourt & Baroni (2018), demonstrates that agents can achieve high accuracy in REF games by conveying low-level details about the target. A comparison of the CtD outcomes for the THING and QRC datasets in Table 4 further underscores this observation. Interestingly, these datasets encode the same information in contrasting manners. While representations of the THING data is entirely compositional, that of the QRC data is not. Consequently, although accuracy is high for both datasets, 1.0 and 0.956 respectively, the CBM for the THING dataset is perfect (1.00), whereas for the QRC dataset, it tends toward randomness (0.172).

Moreover, the perfect AMI score for the QRC game highlights its inadequacy in evaluating compositionality in such datasets. Table 6 describes a concrete example for this phenomenon, extracted from the experiment reported in Table 4. As seen, the sender in the game generates two entirely distinct messages, comprising mutually exclusive words, for two phrases that vary in only a single concept (`ceramic` Vs. `leather`).

Table 6: Non-compositional messages for the QRC game. Changing a single concept in a phrase results in changes to all words in a message.

| Phrase | Gold concepts | Generated words |
|---|---|---|
| circle.red.massive.classic.ceramic | 1.11.29.35.46 | 37.1.32.43.38 |
| circle.red.massive.classic.leather | 1.11.29.35.48 | 13.27.25.34.36 |

## D  LEARNING WITH MULTIPLE TARGETS

In this work, we argue that multiple-targets setting is essential for compositionality to emerge. We report results for the varied-targets experiment in Section 5.2. To further support the importance of the multi-target setting for the emergence of compositionality, we include results from experiments on additional datasets, as well as multi-target experiments conducted for the COMPOSE training phase.

### D.1  MULTIPLE TARGETS DURING DECOMPOSE

Figure 11 presents additional results from the varied-targets experiments on the SHAPE and MNIST datasets, as discussed in Section 5.2. As shown, reducing the number of targets led to a significant decrease in all compositionality metrics across all six configurations.

Interestingly, decreasing the number of targets on the SHAPE dataset increases accuracy (orange line) when GS and QT communication are in used. We attribute this phenomenon to the model's tendency to describe the target using low-level features when a smaller number of targets is used. This phenomenon is less pronounced in the MNIST dataset, probably due to the shorter phrases used (2 concepts) compared to the 4-concept phrases used in the SHAPE dataset. With CB communication, the model is trained to optimize both task and commitment losses (See Section 3 for details). Consequently, an inherent dependency exists between accuracy and compositionality, making it more challenging to analyze the results of one independently from the other.

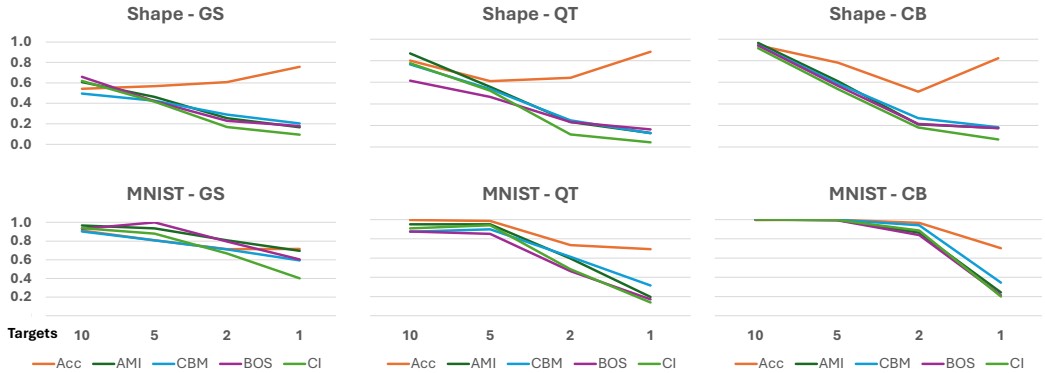

Figure 11: Learning concepts requires multiple targets. Illustrating results of the DECOMPOSE step on the SHAPE (Top) and the MNIST (Bottom) datasets.

### D.2  MULTIPLE TARGETS DURING COMPOSE

While multiple targets are indispensable for the DECOMPOSE step, the classical REF game relies on a single target. The use of a single target requires less supervision, making it more appealing for studying emergent communication. Table 7 presents a comparison of results between communicating with 20 targets versus a single target during the **COMPOSE** step. In the SHAPE and MNIST games, there is a moderate decline in performance, which we attribute to differences in representing a single image versus a collection of images sharing the same concepts. Refer to Appendix C.5 and Figure 9 for more details on the image differences in the MNIST dataset. Conversely, in the THING and QRC datasets, objects are uniquely identified, so a single and multiple images have the same representation,

Table 7: Comparing performance of single vs multiple targets during the COMPOSE step of the MREF game.

| Dataset | Target(s) | l | #p | #m | #w | ACC | AMI | POS | BOS | CI | CBM |
|---------|-----------|---|-----|-----|-----|-------|-------|-------|-------|-------|-------|
| Thing | 20 | 5 | 958 | 958 | 50 | 1.000 | 1.000 | 0.033 | 0.975 | 1.000 | 1.000 |
| | 1 | 5 | 958 | 958 | 50 | 1.000 | 1.000 | 0.032 | 0.975 | 1.000 | 1.000 |
| Shape | 20 | 4 | 34 | 192 | 17 | 0.988 | 0.745 | 0.045 | 0.841 | 0.999 | 1.000 |
| | 1 | 4 | 34 | 489 | 17 | 0.696 | 0.696 | 0.036 | 0.606 | 0.709 | 0.883 |
| MNIST | 20 | 2 | 15 | 32 | 10 | 0.710 | 0.895 | 0.210 | 0.279 | 0.828 | 0.902 |
| | 1 | 2 | 15 | 72 | 10 | 0.662 | 0.628 | 0.083 | 0.163 | 0.552 | 0.782 |
| COCO | 20 | 2 | 26 | 97 | 56 | 0.654 | 0.805 | 0.283 | 0.295 | 0.342 | 0.638 |
| | 1 | 2 | 26 | 322 | 79 | 0.922 | 0.426 | 0.195 | 0.173 | 0.117 | 0.357 |
| QRC | 20 | 5 | 958 | 958 | 50 | 0.956 | 1.000 | 0.001 | 0.006 | 0.013 | 0.172 |
| | 1 | 5 | 958 | 958 | 50 | 0.984 | 1.000 | 0.001 | 0.008 | 0.012 | 0.167 |

and thus no differences in performance are observed. In the COCO dataset, images differ vastly from each other even when sharing the same concepts (see Figure 10). Thus, the representation of a single image significantly differs from creating a representation by averaging multiple images. While accuracy for a single image game variant (0.922) is notably higher than accuracy for the 20-targets variant (0.654), AMI, CI and CBM (0.805, 0.342 and 0.638, respectively) for the 20-target variant are much higher than those for the single target (0.426, 0.117 and 0.357, respectively).

## E    CODEBOOK AS A DISCRETE COMMUNICATION CHANNEL

In this section, we offer additional details on the two codebook variants employed in our experiments. We begin with a formal definition of the codebook used, followed by a comparison of the results from these two codebook configurations on the MNIST dataset.

Formally, we define a discrete latent codebook $CB \in \mathbb{R}^{W \times d}$ where $W = |w|$ is the number of words in the codebook and $d$ is the dimension of each word $w_i$. The discrete latent variables $\tilde{w}$ are then calculated by a nearest neighbour lookup over the codebook: $\tilde{w} = q(z) = w_k$ where $k = \arg\min_j ||z - w_j||^2$, and $z = z_\theta(u_\theta(x))$ is the latent vector generated by the sender's network for a given data sample $x$. We refer to this plain codebook setup as **P-CB**.

We follow a recent work (Zheng & Vedaldi, 2023) that suggests a re-initialization method to encourage the usage of all words in the codebook. We refer to codebook with re-initialization as **C-CB**. Defining $N_k^{(t)}$ to be the average usage of code vectors in each training mini-batch, it is updated as

$$N_k^{(t)} = N_k^{(t-1)} \cdot \gamma + \frac{n_k^{(t)}}{B} \cdot (1 - \gamma), \tag{9}$$

where $n_k^{(t)}$ is the number of encoded samples $z_k$ that will be quantized to the closest codebook entry $w_k$, $B$ is the number of samples in a batch and $\gamma$ is a decay hyper-parameter which we set to 0.99.

Code words are then reinitialized based on their usage. We follow the probabilistic random selection suggested by Zheng & Vedaldi (2023) for sampling an anchor set from a batch. Specifically, we consider a probability $p = \frac{\exp(-D_{i,k})}{\sum_{i=1}^{B} \exp(-D_{i,k})}$, where $D_{i,k}$ is the distance between $z_i$ and $w_k$. Our goal is to reinitialize less-used code words while refraining from modifying used ones. For that we define a decay factor $\alpha_k$ calculated for each of the code words,

$$\alpha_k^{(t)} = \exp^{-N_k^{(t)} W \frac{\delta}{1-\gamma} - \epsilon} \tag{10}$$

where $\epsilon$ is a small constant to ensure the entries are assigned with the average values of words along different batches, $\delta$ is a constant which we set to 10 as in (Zheng & Vedaldi, 2023), and $K$ is the number of words in the codebook. The update is then done by

$$w_k^{(t)} = w_k^{(t-1)} \cdot (1 - \alpha_k^{(t)}) + \hat{z}_k^{(t)} \cdot \alpha_k^{(t)}, \tag{11}$$

Table 8: A comparison of the performance between a standard codebook (P-CB) and a re-initialized codebook aimed at reducing dead codes (C-CB) on the MNIST dataset.

| Codebook | Size | Acc | AMI | POS | BOS | CI | CBM | Umbig | Para |
|----------|------|-----|-----|-----|-----|-----|-----|-------|------|
| P-CB | 10 | 0.999 | 1.000 | 1.000 | 1.000 | 1.000 | 1.000 | 0 | 0 |
| P-CB | 15 | 0.339 | 0.664 | 0.987 | 0.980 | 0.627 | 0.399 | 0.601 | 0 |
| P-CB | 20 | 0.999 | 1.000 | 1.000 | 1.000 | 1.000 | 1.000 | 0 | 0 |
| C-CB | 10 | 0.999 | 1.000 | 1.000 | 1.000 | 1.000 | 1.000 | 0 | 0 |
| C-CB | 15 | 1.000 | 0.936 | 0.882 | 0.762 | 0.840 | 0.840 | 0 | 0.160 |
| C-CB | 20 | 0.999 | 0.885 | 0.801 | 0.675 | 0.665 | 0.665 | 0 | 0.335 |

where $\hat{z}_k^{(t)} \in \hat{Z}^{K \times z_q}$ is the sampled anchor. We set hyper-parameters and constants according to the recommendations in Zheng & Vedaldi (2023).

### E.1 OVER-COMPLETE CODEBOOK

Table 8 compares the performance of a plain codebook (P-CB) with that of a codebook employing the re-initialization method (C-CB) proposed by Zheng & Vedaldi (2023). The results are evaluated for both complete and over-complete codebooks.

As observed, increasing the C-CB codebook size does not reduce task performance but does decrease the CBM by introducing more paraphrases, with scores of 0.16 and 0.335 for the 15 and 20 codebook words, respectively. This effect is primarily due to the C-CB algorithm, which continuously re-initializes dead codes. To further evaluate this, we conducted experiments using the same configuration with the original CB code (P-CB). As seen, the codebook initialization for the 15-word size was unsuccessful, leading to low accuracy (0.339) and the usage of only 4 codes, with a high ambiguity rate (0.601). In contrast, for a codebook size of 20, the initialization was successful, resulting in perfect accuracy (1.000) and optimal scores across all compositionality metrics.

### E.2 CODEBOOK UTILIZATION

Table 9 presents the utilization of the two codebooks (P-CB and C-CB), initialized with 15 and 20 random words. The results, evaluated using CBM on a test set of 1,000 samples, reflect only successful matches, with the remaining cases classified as ambiguous or paraphrased, as detailed in Table 8. Notably, P-CB with 20 code words demonstrates $100\%$ utilization for the presented words, indicating that the remaining 10 words are "dead code" that were never used.

## F VARIED MESSAGE LENGTH ANALYSIS

In this section, we provide a detailed analysis of the impact of message length on accuracy and compositionality metrics, for both CB-based and RNN-based communication protocols. We highlight the limitations of RNN-based protocols in handling variable-length messages and propose several potential improvements for CB-based communication. The implementation of these improvements is left for future research.

### F.1 THE IMPACT OF MESSAGE LENGTH ON ACCURACY AND COMPOSITIONALITY

In Figure 12 we compare accuracy and compositionality metrics across the four compositional datasets for the three communication protocols. For CB-based communication, both accuracy and compositionality increase significantly as message length grows, reaching an optimal point that matches the phrase length. Beyond this point, both metrics decline. In contrast, for RNN-based communication, accuracy continues to improve with increasing message length (except for GS on THING), while compositionality metrics either decline or stay unchanged. The variations in accuracy behavior underscore the limitations of the RNN mechanism in identifying an optimal message length. As further seen in Table 10, both GS and QT typically make use of the maximum allowed message length. This aligns with our observation that a high-capacity channel enables better accuracy by

Table 9: Assessing codebook utilization using the CBM metric on $1,000$ samples from the MNIST test set. The results pertain to the DECOMPOSE phase for codebooks containing 15 and 20 code words.

| | Zero | One | Two | Three | Four | Five | Six | Seven | Eight | Nine |
|---|---|---|---|---|---|---|---|---|---|---|
| **Ground Truth** | 96 | 90 | 100 | 109 | 94 | 112 | 103 | 103 | 102 | 91 |
| **P-CB with 15 code words** | | | | | | | | | | |
| Index | n/a | n/a | 1 | n/a | 3 | 2 | n/a | 0 | n/a | n/a |
| Util (#) | 0 | 0 | 90 | 0 | 94 | 112 | 0 | 103 | 0 | 0 |
| Util (%) | 0 | 0 | 90 | 0 | 100 | 100 | 0 | 100 | 0 | 0 |
| **P-CB with 20 code words** | | | | | | | | | | |
| Index | 8 | 1 | 3 | 5 | 6 | 7 | 4 | 0 | 2 | 9 |
| Util (#) | 96 | 90 | 100 | 109 | 94 | 112 | 103 | 103 | 102 | 91 |
| Util (%) | 100 | 100 | 100 | 100 | 100 | 100 | 100 | 100 | 100 | 100 |
| **C-CB with 15 code words** | | | | | | | | | | |
| Index | 11 | 1 | 8 | 5 | 6 | 7 | 4 | 0 | 2 | 12 |
| Util (#) | 96 | 90 | 63 | 91 | 94 | 64 | 103 | 103 | 75 | 61 |
| Util (%) | 100 | 100 | 63 | 83 | 100 | 57 | 100 | 100 | 74 | 67 |
| **C-CB with 20 code words** | | | | | | | | | | |
| Index | 14 | 1 | 9 | 5 | 6 | 8 | 4 | 7 | 2 | 15 |
| Util (#) | 54 | 90 | 55 | 95 | 71 | 65 | 66 | 54 | 65 | 50 |
| Util (%) | 56 | 100 | 55 | 87 | 76 | 58 | 64 | 52 | 64 | 55 |

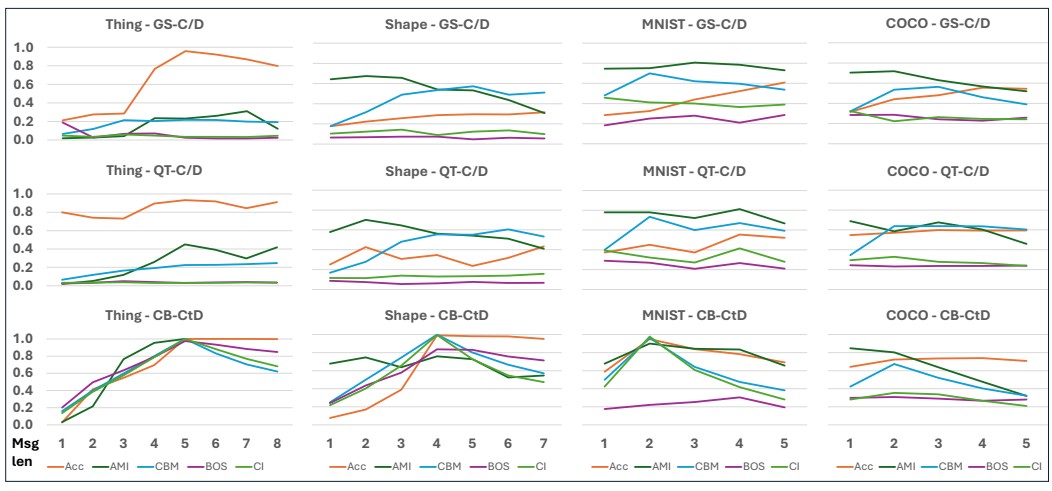

Figure 12: An evaluation of accuracy and compositionality metrics (Y-axis) across varied message lengths (X-axis) for the three communication protocols (GS, QT, CB) on the four compositional datasets (THING, SHAPE, MNIST, and COCO).

conveying low-level details. This unintended phenomenon indicates that additional regularization is necessary for the RNN to foster compositionality. In this work, to enable a fair comparison, we set the maximum message length to match the length of the phrases in the dataset for all communication protocols. Figure 12 presents CtD results for CB, but C/T results for GS and QT, as the latter generally performs better than CtD for these communication protocols.

## F.2 CB-BASED VARIED-LENGTH CONTROL

Several alternatives can be considered for the oracle-based length configuration currently utilized by the CB protocol. First, message length could be treated as an additional hyperparameter, with its optimal value determined on a validation set. This method should perform well, as suggested

Table 10: A comparison between the potential and actual average message length for the two RNN-based communication protocols (GS and QT). Bold font highlights results where the message length matches the phrase length of the datasets.

| Msg len | THING | | SHAPE | | MNIST | | COCO | |
|---|---|---|---|---|---|---|---|---|
| | GS | QT | GS | QT | GS | QT | GS | QT |
| 1 | 1.000 | 1.000 | 1.000 | 0.998 | 1.000 | 0.979 | 1.000 | 1.000 |
| 2 | 2.000 | 1.996 | 1.996 | 2.000 | **1.515** | **1.882** | **1.938** | **2.000** |
| 3 | 3.000 | 2.818 | 3.000 | 2.979 | 1.992 | 2.885 | 3.000 | 3.000 |
| 4 | 4.000 | 3.965 | **3.905** | **3.961** | 3.998 | 3.526 | 4.000 | 4.000 |
| 5 | **5.000** | **4.919** | 4.952 | 4.618 | 4.860 | 4.612 | 5.000 | 5.000 |
| 6 | 6.000 | 5.870 | 5.902 | 5.988 | | | | |
| 7 | 7.000 | 6.938 | 6.939 | 6.964 | | | | |
| 8 | 8.000 | 7.771 | | | | | | |

by the CB results in Figure 12, provided the lengths of the phrases are uniform. However, a more sophisticated mechanism is required when the dataset contains varied-length phrases. One possible approach is to train a similarity threshold that learns to dynamically extract varying numbers of words from the codebook. We leave the exploration of this and other varied-length control mechanisms for future research.

## G    CONFIGURATION AND TRAINING DETAILS

We run all experiments over a modified version of the Egg framework (Kharitonov et al., 2019).[1] In our version the communication modules, $z_\theta$ and $z_\phi$, used by the sender and the receiver, respectively, are totally separated from their perceptual modules, which we term 'agents'. See schematic illustration in Figure 2. This separation allow us to use the exact same communication modules for different games. The Gumbel-Softmax (GS), Quantized (QT), and coodbook-based (CB) protocols are implemented at the communication layer. For GS we use an implementation provided by the Egg framework, where we do not allow temperature to be learned, and set the straight-through estimator to False. For QT protocol we followed parameter recommendations from Carmeli et al. (2023) and use a binary quantization in all experiments. For CB we adapted the implementation code provided by Zheng & Vedaldi (2023) and a code from Oord et al. (2018).[2] Both GS and QT uses a recurrent neural network (RNN) for generating multiple words within a message. Refer to Table 11 for details on specific RNN and other hyper-parameters. The CB-based protocol takes the message length as a parameter, which we set to 1 for the DECOMPOSE step and to the maximum phrase length, as determined by the dataset, for the COMPOSE step. Refer to column 'l' in Table 4 for the exact message-length value for each dataset.

Table 11: Agents' hyper-parameters used for obtaining the results in Table 4.

| Batch Size | lr | Sender Targets | Sender Distr | Receiver Targets | Receiver Distr | Cell Type | Sender Hidden | Sender Embed | Receiver Hidden | Receiver Embed |
|---|---|---|---|---|---|---|---|---|---|---|
| 10 | 0.0005 | 20 | 0 | 20 | 20 | LSTM | 100 | 500 | 100 | 500 |

The communication protocols further differ in their vocabulary-size settings. We aim to set this size, $v$, to match the number of basic concepts in the data, which is dataset dependent. For GS $v$ is equal to word length, $d$, which we set to match the number of concepts, $c$ in the dataset. For QT $v = 2^d$. We set $d$ such that $v \geq c$. For CB, vocabulary size, $v$, is equal to the number of vectors in the codebook, which we set to match $c$, the number of concepts in the dataset. See columns 'd' and '#c' in Table 4 for the actual values of these parameters.

Agent architectures, $u_\theta$ and $u_\phi$, vary slightly across games to best encode the input $X$ into latent vectors $U^s$ and $U^r$. Specifically, for the THING game we employed two fully connected feed-forward

---

[1] Available at `https://github.com/bcarmeli/egg_qtc`.
[2] Available at `https://github.com/airalcorn2/vqvae-pytorch`.

layers with input dimension of $d = 270$. Objects in that game are encoded by concatenating five one-hot vectors, one per attribute. For the SHAPE, MNIST and QRC games we used four-layers convolutional neural network, similar to Mu & Goodman (2021). For the COCO game we adopted the Resnet50 (He et al., 2016) pretrained architecture to embed images into the feature space, followed by two fully connected feed-forward layers.

**Codebook initialization**  Proper codebook initialization is crucial for optimal performance in both the DECOMPOSE and COMPOSE phases. Initializing the codebook with too similar vectors leads to difficulties in associating different concepts with distinct words, while initializing with vectors that are too dissimilar results in codebook underutilization. We observed that adopting the online clustering approach proposed by Zheng & Vedaldi (2023) significantly enhances codebook performance. Moreover, the quality of codebook initialization is the sole distinction between the C/D and CtD results, obtained during the COMPOSE step. As demonstrated in Table 2, this discrepancy profoundly influences agents' performance.

**Codebook training**  In the original VQ-VAE work (Oord et al., 2018), the authors discuss two types of loss: commitment loss ($||\text{sg}[z] - w||_2^2$) and dictionary loss ($||z - \text{sg}[w]||_2^2$). In an appendix to the paper, they propose using an exponential moving average (EMA) instead of dictionary loss. After experimenting with both approaches, we found that EMA outperforms dictionary loss in all our setups.

In a subsequent study on online clustering (Zheng & Vedaldi, 2023), the authors utilize EMA to re-initialize inactive code vectors, a process conducted independently of the commitment and dictionary loss formulas. Our experiments revealed that with re-initialization, using both the dictionary and commitment losses is more effective than substituting dictionary loss with its EMA counterpart. Therefore, we adopt the loss and re-initialization methods proposed by Zheng & Vedaldi (2023) across all experiments.

We tested various values of $\beta_2$, which balances the commitment and dictionary losses, ranging from 0.1 to 1.0. After finding no significant impact on results with different values, we set it to 1.0 for all experiments..

**Early stopping**  We conduct experiments over multiple epochs and choose the checkpoint exhibiting the lowest validation loss. Importantly, the CtD optimizes a trade-off between the task loss and the codebook loss. We use $\beta_1$ hyper-parameter to balance this trade-off. After testing values ranging from 0.25 to 1.0 and observing no significant differences, we set $\beta_1$ to 1.0 for all experiments. Refer to Section 3.3 for more details. We note instances where these losses are in conflict. Refer to Figure 6 for examples. Although we consistently select the checkpoint with the lowest **combined** loss across all experiments, choosing the checkpoint that minimizes the task loss yields improved accuracy, while selecting the checkpoint that minimizes the commitment loss enhances compositionality.

We run all experiments on a single A100 GPU with 40 GB of RAM. Model size is less than 10MB. We use the 'Adam' optimizer with learning rate as specified in Table 11. We usually trained the model for 200 epochs which took about 10 hours to complete. Our code is available at `https://github.com/bcarmeli/egg_qtc`. Please contact the authors via email for access.

## H  COMPARISON WITH EARLIER WORKS

This section describes additional related works and offers a more detailed comparison with two specific lines of research.

### H.1  LEARNING WITH MULTIPLE LOSSES

Several works identify the inherent accuracy-compositionality trade-off. Tucker et al. (2022) identify this as a trade-off between complexity and expressiveness. Guo et al. (2021) study the trade-off between contextual complexity and unpredictability. Rita et al. (2022) observe that the loss can be decomposed into an information term and a co-adaptation term, which stems from the listener's difficulty to cooperate with the sender. Two-term losses are inherent to codebook-based architectures and can naturally map to the the two complementary tasks. While achieving high

accuracy requires minimizing the task loss, achieving high compositionality requires minimizing the codebook commitment loss.

## H.2 TWO-STEP TRAINING

Recent work on compositionality in neural networks highlights meta-learning (Finn et al., 2017) as key to out-of-distribution generalization. Russin et al. (2024) proposes that compositional generalization can be achieved using two training loops: the inner-loop generalizes o.o.d. due to the inductive biases from the outer-loop. Lake & Baroni (2023) show that transformers can achieve compositional generalization through in-context learning (ICL) in the inner-loop, after training the outer-loop on next-word prediction. While technically different from CtD, these insights support our results. In CtD, the inner-loop learns concepts, and the outer-loop composes complex phrases using them.

## H.3 COMPARISON WITH OBVERTER

A study by Bogin et al. (2018) introduces a Context-Consistent Obverter architecture, demonstrating the emergence of discrete communication from images. They report both accuracy and compositionality, measured using CI, and compare their results with earlier Obverter findings from Choi et al. (2018). Both studies use a dataset similar to our SHAPE dataset. Their version consists of 8 shapes and 5 colors, with the shapes in 3D.

Their dataset contains $8 \times 5 = 40$ unique NL phrases, each consisting of a shape and a color. In contrast, our dataset has $6 \times 5 \times 3 \times 3 = 270$ unique NL phrases, with each phrase being 4 concepts long. Refer to Appendix B for details. Achieving high compositionality scores in our setup is, therefore, more challenging.

Additionally, in Bogin et al. (2018) and Choi et al. (2018) setups, the receiver observes a single distractor, while in our setup, the receiver deals with 20 distractors, making the random baseline in their setup (0.5) significantly higher than ours (0.05). Furthermore, their data split allows the same phrase to appear in both the training and test sets, making it easier for agents to perform well on the test set.

Despite these differences, which simplify the task in these studies, Bogin et al. reported a CI score of $0.4$, while Choi et al. only demonstrated the potential segmentation of emergent communication (EC) messages into natural language (NL) concepts. In contrast, by using the CtD method, we achieved perfect accuracy and compositionality scores, even on the more complex task, which includes 20 distractors and a more challenging train/test split.

## H.4 COMPARISON WITH VQ-VIB

A study by Tucker et al. (2022) is the only one we know of that suggest to use a discrete codebook in emergent communication. Their usage of a codebook is significantly different than ours. In their work, Tucker et al. explore the trade-off between complexity, measured by the number of bits allocated for communication, and informativeness, which corresponds to accuracy in solving a referential task. They demonstrate that a balance between complexity and informativeness can be achieved using a discrete codebook, referred to as VQ-VIB.

Notably, Tucker et al. do not address the compositionality of the emergent communication, nor do they provide any compositionality measurements, which is a central focus of our work. Additionally, the dataset they used consists of single-color phrases that do not require compositionality to be solved.

Moreover, Tucker et al. employ a variational auto-encoder (VAE) to learn a distribution over input images, whereas our approach uses a deterministic auto-encoder to structure the codebook. While they suggest a method for extracting multiple vectors from the codebook by segmenting the input vector $U^s$ and finding the nearest vector for each segment, our approach instead identifies the $l$ code-vectors most similar to $U^s$.

In summary, despite both CtD and Tucker et al. (2022) using VQ-VAE-based codebooks, they do so in fundamentally different ways and for distinct purposes.

## I   A WIDER VIEW OF THE ACCURACY COMPOSITIONALITY TRADE-OFF

In this section, we refer to few works related to the accuracy-compositionality trade-off aspects highlighted in the main body of the paper. Additionally, we reference recent studies that address similar aspects in more conventional setups and suggest promising future directions.

**Channel capacity and information Bottleneck**   Broadly speaking, a wide channel aids agents in achieving high accuracy. For instance, with continuous communication (see Section A.2), which practically offer unlimited capacity, agents are able to achieve perfect accuracy in REF games (Carmeli et al., 2023). Conversely, the information bottleneck (Slonim, 2002; Kharitonov et al., 2020) serves as the primary inductive bias assumed by the EC community for the emergence of natural language traits. Hence, an inherent trade-off exists between accuracy and compositionality objectives.

**Discrete channel**   As a somewhat orthogonal line of thought, the EC community assumes that discrete communication is an essential bias for the emergence of natural language traits (Lazaridou et al., 2017; Choi et al., 2018; Havrylov & Titov, 2017). This assumed bias, which goes beyond simply adding an information bottleneck (Koh et al., 2020), makes the most sense as the most notable differences between natural language and neural networks relate to this aspect.

**Measuring compositionality**   Adding to these challenges, until recently (Carmeli et al., 2024), a straightforward and intuitive metric for assessing compositionality was lacking (Hupkes et al., 2020; Dankers & Titov, 2023; Andreas, 2019). Consequently, evaluating the impact of channel capacity and discreteness on accuracy and compositionality posed significant challenges.

**Common communication and optimization methods**   Previous studies have proposed various strategies for managing the trade-off between accuracy and compositionality within the discrete communication constraint (Lazaridou et al., 2017; Havrylov & Titov, 2017; Carmeli et al., 2023). The most common approaches (GS and RL) typically operate under the assumption of a low-capacity, discrete channel. Alternatively, methods like quantized communication (Carmeli et al., 2023) aim to address the discreteness constraint through quantization, thereby enabling a wide yet discrete channel. While these methods often achieve respectable levels of accuracy or compositionality, they struggle to improve both simultaneously.

**Variational auto-encoders (VAE)**   Another approach involves the use of auto-encoder architectures (Vincent et al., 2008), typically within the RECON game framework (see Section A.1), to establish the necessary information bottleneck. By leveraging the Beta-VAE (Higgins et al., 2017) and similar architectures (Tucker et al., 2022), compositionality is believed to be attained through the disentanglement of the latent space. These methods, often referred to as *features-as-directions*, operate under the assumption that the representation of concepts can be aligned with dimensions in the latent space. In contrast, the VQ-VAE architecture (Van Den Oord et al., 2017) employ a discrete codebook, aiming to represent concepts directly as points in latent space in a so-called *features-as-points* embedding.

**Superposition and feature-as-directions**   Recent advancements in mechanistic interpretability (Olah, 2023; Elhage et al., 2022) have shed light on the challenge of disentangling the latent space, revealing that superposition is an inherent phenomenon in neural networks (Bricken et al., 2023). To address this issue, some researchers propose the use of sparse auto-encoder (SAE) architectures (Geadah et al., 2018) to learn a dictionary of meaningful features. However, these efforts have achieved only limited success thus far. Notably, SAE architectures compromise the information bottleneck assumption by assuming a wide but sparse information channel. We are not aware of any work that studies such architecture under the EC setup.

**Codebook and feature-as-points**   A notable recent study by (Tamkin et al., 2023) incorporates a discrete codebook into the transformer architecture. They demonstrate that with such a discrete bottleneck, interpretable features can be represented by combining a number of code words. Moreover, they offer an analysis comparing the distinctions between *features-as-points* and *features-as-directions*, favoring the former. We find this comparison particularly relevant to our work and to emergent communication architectures in general.

**Conclusion**   In conclusion, we argue that learning to communicate while utilizing a codebook, as done in the CtD approach, enjoys the advantages of *features-as-points* explained by (Tamkin et al., 2023), while VAE-based architectures, which seek for a *features-as-directions* representation, suffer from the less-compositional *feature-as-directions* representation.

## J   CTD LIMITATIONS AND FUTURE WORK

Several aspects of the CtD approach warrant further investigation.

**Self-supervision**   While the original REF and RECON games require no supervision, the DIFF and MREF multi-target games necessitate Oracle supervision to identify targets sharing similar concepts. This supervision directs the network in extracting concepts of interest to the Oracle. Hence, exploring methods for enabling the sender to identify similar images without Oracle supervision is an important improvement that require additional research.

**Dynamic word allocation**   Although we initially aligned the size of the codebook with the number of concepts in the vocabulary (see Appendies E and G for specifics), there is potential for a dynamic word allocation strategy. In this approach, the codebook itself would autonomously determine whether to initiate a new code or utilize an existing one to describe a potentially novel concept. Exploring algorithms such as those proposed by Ghahramani & Griffiths (2005), which facilitate the dynamic allocation of new vectors for emerging concepts, presents a promising avenue for future research.

**Varied-length control**   Unlike the GS and QT protocols, which support communication of varied-length messages, the CB protocol requires the message length as an input. Specifically, this input determines the number of vectors to be extracted from the codebook. In Appendix F we discuss this limitation in details. Investigating a more flexible configuration for vector extraction based on similarity thresholds would be valuable.

**Repeated words**   CB-based communication generates bag-of-words messages, where identical concepts cannot be repeated within the same message. This limitation is not an issue when describing a single object in datasets like THING or SHAPE, as repetition (e.g., mentioning the concept 'Blue' twice) is unnecessary. However, in cases such as an MNIST image depicting the number $11$, the need arises to repeat the concept '1' twice to accurately describe the two identical digits. Although this might appear to be a limitation of the CB-based protocol, repetition is, in fact, necessary for describing multiple instances of an object, as required by datasets like MNIST and COCO. RNN-based approaches also lack a built-in mechanism for "sequential sampling without repetition," which can lead to unnecessary repetition of concepts when describing a single object. To the best of our knowledge, this issue has not been addressed in previous research, and we propose it as a topic for future investigation.

**Index-based Communication**   In the proposed CB communication approach, the sender transmits the full latent vector of a concept to the receiver. In contrast, words in natural language function as indices pointing to the latent space. It is feasible to send the index of a word from the codebook instead of the entire vector, provided the receiver can map these indices back to their corresponding meanings. This index-based communication would require the receiver to have access to the codebook. Addressing this potential enhancement is left for future research.

