# OpenReview forum: "CtD: Composition through Decomposition in Emergent Communication"
_ICLR.cc/2025/Conference — ICLR 2025 Poster_

### Official Review · Reviewer_LvJV · 2024-10-28

**Soundness:** 3
**Presentation:** 3
**Contribution:** 3
**Rating:** 6
**Confidence:** 4

**Summary:**

The paper studies how to improve the model’s compositional generalization ability. Specifically, the authors propose a two-stage training pipeline, in which the model first learns to decompose an image into basic concepts using a codebook and then recombines the learned features to describe novel concepts. The experiments on different image datasets showcase that the proposed method indeed improves the model’s ability compared with other baselines. However, the proposed method needs a carefully selected training set, which limits its applicability to more practical scenarios. Also, the paper hasn’t discussed an important line of work related to this problem, i.e., iterated learning, which is a multi-generation-based training pipeline that can amplify the inherent bias towards compositional mappings automatically. I guess combining the method proposed in this paper and multi-generation training can further improve the model’s ability on compositional generalization. In summary, I think the paper has the potential, but due to the reasons above, I would like to give a negative score for the current version.

**Strengths:**

- The presentation is good, and the paper is easy to follow.
- The experiments are well conducted, covering both toy data and real images.
- The conclusions drawn from the experiments are inspiring. (E.g., I like the results in Figure 3.)
- The zero-shot results are interesting.

**Weaknesses:**

- The paper doesn't mention the multi-generation-based approach for the compositional generalization problem. Actually, the two-stage training proposed here can also be analyzed using similar theoretical tools provided in related works, e.g., [3]. Furthermore, [1] also studies the compositional generalization problem in the context of emergent communication, which is identical to the one studied in this paper. Such an iterated training fashion has proven to be effective in many other fields, e.g., molecular graph prediction [3], visual VQA [2], large VLM like CLIP [4], etc. It might be helpful to discuss (or even compare) with methods in this line of work. So, providing more discussions about the potential synergies or tradeoffs between the decomposition-composition approach and multi-generational training would make the paper stronger.
- The discussions and analysis throughout the paper are high-level. A more in-depth discussion of the dataset selection design in different phases would make the paper stronger. For example, the authors mentioned in line 190 that “In the DECOMPOSE step all targets share a phrase composed from exactly one FVP”, what will happen if this assumption is violated? How will it influence the final results? Will the network be robust enough when this assumption is mildly violated? Since this assumption is too strong to be true in many practical scenarios, relying too much on this assumption might harm the contribution of the paper. I believe conducting ablation studies where the single-FVP assumption is systematically relaxed to varying degrees would make the paper stronger.

[1] Ren, Yi, et al. "Compositional languages emerge in a neural iterated learning model." ICLR 2020

[2] Vani, Ankit, et al. "Iterated learning for emergent systematicity in VQA." ICLR 2021

[3] Ren, Yi, et al. "Improving compositional generalization using iterated learning and simplicial embeddings." NeurIPS 2023

[4] Zheng, Chenhao, et al. "Iterated learning improves compositionality in large vision-language models." CVPR 2024

**Questions:**

Please refer to the weakness part.

---

> ### Author Response · Authors · 2024-11-19
> **Respnose to reviewers LvJV (1)**
>
> We sincerely thank the reviewers for their thorough and detailed feedback, and for their positive assessment of the presentation of our work, the comprehensiveness of our experimental setup, and the inspiring nature of our conclusions. We would like to emphasize that the zero-shot results, which the reviewers found particularly interesting, are a direct consequence of the agents' ability to decompose data into compositional concept representations. We acknowledge that Iterated Learning (IL) can be viewed as a complementary approach to CtD framework. We added a paragraph discussing it to the related work section (6) in the main body of the paper. We also provide further insights regarding the single-FVP assumption below.
> In light of these points, we respectfully request that the reviewers reconsider their final evaluation in a more favorable light.
>
> ### [Weakness-1 - IL]:
> We appreciate the reviewer for highlighting the iterated learning (IL) framework. In response, we have added a dedicated paragraph discussing this line of work in the related work section. While IL is indeed relevant to compositionality, we view it as complementary to CtD, as IL largely addresses the refinement of compositional structures rather than the emergence of new concepts, which is the focus of CtD. CtD specifically explores how these concepts originate, with its strengths most evident in the zero-shot results, which demonstrate that compositional representations can be learned within a single iteration. We acknowledge the potential synergy between these approaches and agree that integrating them could lead to further advancements in this field.
>
> ### [Weakness-2 - Discussion is High-Level]:
> > “In the DECOMPOSE step all targets share a phrase composed from exactly one FVP”
>
> We consider the decomposition phase to be central to the "Compose through Decompose" (CtD) approach. In our methodology, we deliberately separate the processes of concept learning (decomposition) and concept usage (composition). In this paper, we hypothesize that learning a dedicated representation for a specific concept cannot be effectively achieved in simple referential games with a single target. Instead, we demonstrate that the presence of multiple targets sharing a concept is necessary, and that sharing exactly one concept leads to an optimal learning process.
> A detailed discussion on the necessity of multi-targets for the concept learning process is provided in Appendix D.
> In Appendix F, we further analyze the scenario where objects share several FVPs, and the sender is constrained to transmit only a subset of these features due to controlled message length.
>
> > “what will happen if this assumption is violated? How will it influence the final results? Will the network be robust enough when this assumption is mildly violated?”
>
> As demonstrated in prior studies and further illustrated in our analysis of Composition without Decomposition (C/D) performance (lines 397-409), as well as in the rows marked with C/D in Table 4, compositionality performance deteriorates when targets share multiple FVPs. While a network with sufficient capacity can generalize to these cases and maintain high accuracy, its ability to construct a compositional representation is significantly reduced.
>
> > “I believe conducting ablation studies where the single-FVP assumption is systematically relaxed to varying degrees would make the paper stronger.”
>
> As previously mentioned, the experiments in which we run the composition phase without the preceding decomposition phase (C/D), and while controline the message length (Appendix F), serve as an ablation study for the single-FVP assumption.
>
> > “Since this assumption is too strong to be true in many practical scenarios, relying too much on this assumption might harm the contribution of the paper.”
>
> We acknowledge that the ‘single FVP’ assumption requires additional supervision. However, we contend that the supervision needed is not excessively strong, as it only involves tasks like having an agent indicate the similarity of two objects (e.g., pointing to a red triangle and a blue triangle as being "similar"). Furthermore, the multi-target setting is analogous to introducing distractors that share all features except the relevant FVP. It remains to be seen whether this assumption can be relaxed. Some might argue that this notion of "similarity" is somewhat subjective and is not inherently present in the data, necessitating external input. For instance, taxonomic distinctions across languages, such as the numerous words for snow in Inuit, arise not solely from visual differences but from the cultural, environmental, and functional pressures within that context. As such, the requirement for many words—or just a few—depends on task-specific needs, which may require additional supervision. Lastly, as suggested, combining our approach with Iterated Learning (IL) may enable us to relax this assumption, and we are eager to explore this possibility.

---

> > ### Author Response · Authors · 2024-11-25
> > **Request for response**
> >
> > Dear LvJV reviewers,
> >
> > The public discussion phase is nearing its conclusion and will end on November 26th.
> >
> > We kindly ask you to take a moment to review our responses to your questions during this remaining time.
> > If there are any concerns we may have missed or questions that remain unanswered, please do not hesitate to let us know.

---

> > > ### Comment · Reviewer_LvJV · 2024-11-25
> > >
> > > Thanks very much for the author's response, which addresses most of my concerns well. The paper is still a little dense and many interesting discussions are hidden in the Appendix part. So maybe a slight change of the presentation in the next version would be helpful. Anyway, since my major concerns are well addressed, I would increase the soundness and contribution from 2 to 3, and the overall evaluation from 5 to 6. I'm looking forward to seeing the final version of this paper.

---

> > > > ### Author Response · Authors · 2024-11-26
> > > > **Gratitude to Reviewer LvJV for the Second Review Round**
> > > >
> > > > We sincerely thank the reviewers for their thoughtful consideration of our responses.
> > > >
> > > > We believe that incorporating the iterative learning aspect, as suggested, will significantly strengthen the paper.
> > > >
> > > > While working within the constraints of the limited space, we will continue striving to enhance the clarity and presentation of the manuscript.
> > > >
> > > > We greatly appreciate the reviewer’s willingness to reconsider and adjust their scores.

---

### Official Review · Reviewer_Ktng · 2024-10-31

**Soundness:** 2
**Presentation:** 3
**Contribution:** 3
**Rating:** 8
**Confidence:** 2

**Summary:**

This study presents "Composition through Decomposition" (CtD), a method that enables artificial neural agents to achieve compositional generalization, allowing them to describe new images by systematically combining learned basic concepts. CtD involves two stages: first, a “Decompose” step where agents learn to break down complex images into discrete basic concepts using a codebook developed through multi-target coordination games. Then, in the “Compose” step, agents use this codebook to describe novel images by recombining basic concepts, achieving zero-shot generalization in some cases.

**Strengths:**

Originality: Fairly original, builds upon previous work but with somewhat novel contributions

Quality: Very pertinent experiments on a wide range of tasks with a wide range of metrics

Clarity: Sections 1, 2, 4, 5, 6, and 7 are very clear and well written

Significance: Significant contribution to understanding how compositionality can emerge in a learning system, which is an important concept to understand faster and more effective learning

**Weaknesses:**

1. It seems to me that in order to properly do the CtD approach, you need to already know the concepts you want to compose ahead of time in order to setup the 2 phases of training. I don't know how realistic this is in real-world settings.

2. I think the paper should make it clear what the novel part of the work is (from what I understand it is only the 2 phases of training), because now it seems like all 3 parts of section 3 are novel

Potential additional citation:
- https://arxiv.org/pdf/2002.01365

**Questions:**

My questions are based on the weaknesses listed above:

Could the authors discuss potential methods for automatically discovering or learning relevant concepts during the decomposition phase? How might this approach be modified for real-world applications where predefined concepts are not available?

Could the authors add a paragraph in Section 3 or in the introduction that clearly outlines the key innovations of their approach? This would help readers better understand which components are novel and which build upon previous work

---

> ### Author Response · Authors · 2024-11-19
> **Respnose to reviewers Ktng (1)**
>
> We would like to thank the reviewers for their positive feedback, recognizing our work as original, clearly written, and supported by relevant experiments that contribute significantly to the field.
> Below, we address the reviewers' concerns in detail. In light of the positive feedback and our responses to the questions raised, we kindly request the reviewers to reconsider their scores more favorably.
>
> > “Could the authors discuss potential methods for automatically discovering or learning relevant concepts during the decomposition phase? How might this approach be modified for real-world applications where predefined concepts are not available?”
>
> We acknowledge that our approach relies on additional supervision to identify objects that share similar concepts. We argue that the notion of "similarity" is inherently subjective and not naturally present in the data, making external supervision indispensable. For example, taxonomic distinctions across languages, such as the varying number of color terms, stem not only from visual differences but also from cultural, environmental, and functional pressures related to the specific tasks being addressed. It remains an open question whether entirely new concepts can emerge without some form of supervision to indicate that two distinct objects share similar concepts. We referred to that limitation and the potential for utilizing the sender’s representation for a self-supervision in the Limitation section (lines 1791-1795).
> Importantly, our research primarily focuses on the emergence of concepts, aiming to explore the origins of this phenomenon, rather than addressing "real-world applications" where language and concepts are already established. However, representation learning approaches such as Iterated Learning (IL), as suggested by Reviewer LvJV, and discrete world models [1], hold potential for enhancing compositionality in real-world scenarios, and can be applied in addition, or on top of our approach.
>
> [1]  Hafner et al, “Mastering atari with discrete world models” ICLR 2021
>
> > Could the authors add a paragraph in Section 3 or in the introduction that clearly outlines the key innovations of their approach? This would help readers better understand which components are novel and which build upon previous work
>
> We clarified it further by adding a constraining sentence to the intro.

---

> > ### Comment · Reviewer_Ktng · 2024-11-24
> >
> > Thank you for your clarifications and modifications! I have modified my score accordingly.

---

> > > ### Author Response · Authors · 2024-11-26
> > > **Thanks to Reviewer Ktng for the Review**
> > >
> > > We deeply appreciate the reviewers' thoughtful evaluation of our responses and their willingness to reconsider and revise their scores.

---

### Official Review · Reviewer_dKHK · 2024-11-03

**Soundness:** 3
**Presentation:** 4
**Contribution:** 3
**Rating:** 6
**Confidence:** 3

**Summary:**

The paper introduces "Composition through Decomposition" (CtD), an approach enabling artificial neural agents to achieve compositional generalization by breaking down and recomposing concepts. CtD consists of two phases: in the "Decompose" step, agents learn to identify basic concepts in images through a codebook-based communication setup; in the "Compose" step, they use these learned concepts to describe novel images. This structured approach allows for zero-shot generalization to unseen compositions. Experimental results on multiple datasets show CtD can outperform standard methods, achieving high accuracy and compositionality.

**Strengths:**

The approach of "Composition through Decomposition" (CtD) method stands out as a two-step approach where agents first learn to decompose complex objects into simpler concepts before recomposing them. It is an interesting way to get agents to perform compositional inference.

Using a discrete codebook to handle basic concept representations is well-grounded, showing the potential for generalization without additional training. This idea, inspired by linguistic encoding methods, enables efficient compositional representation and surpasses traditional methods in performance on multiple metrics.

The paper evaluates the CtD approach across diverse datasets (e.g., THING, SHAPE, MNIST), demonstrating high accuracy and compositionality scores. The performance evaluation using zero-shot and multi-target settings strengthens the claim that CtD enables compositional generalization.

The authors us multiple metrics (e.g., AMI, CBM, BOS) to give a clear picture of the model's strengths and weaknesses.

I generally thought this paper was well-written and the presentation was good.

**Weaknesses:**

"Several methods have been proposed and evaluated during recent years by the
emergent communication (EC) research community..." -> isn't meta-learning another one to mention here?

"Notably, our method achieves extremely high performance, characterized by perfect accuracy and
compositionality, on multiple datasets." -> just say it achieves perfect accuracy, no need to say extremely high

"we assert that before effectively composing basic concepts into complex ones, agents must acquire the ability
to decompose complex concepts into basic ones" -> must sounds quite strong here. Do you really show that there is no other way to do this? Or is your method one way to accomplish compositionality?

You're using "employed" a lot when really you could just say "use"

I didn't get how the beta of the loss function (game and codebook-tradeoff) was set at first but later it said that the authors "experimented with various values of β1 and β2, ranging from 0.1 to 1.0 and found no significant difference in results." What do you make of this? Why does it not make a big difference? Is this because the loss is dominated by one part in any case?

There is a lot of things going on in Figure 3. Why are there arrows in there? This seems to mostly just show exactly what is in the text already.

I like the ablation studies although it could be a bit better explained why multi-target makes a difference and how the differences between the data sets in this ablation can be explained.

The paper discusses a multi-loss optimization, but the complexity and potential sensitivity of the CtD approach to hyperparameter settings (e.g., β1 and β2 values) could make it less accessible for broader use.

For some datasets, such as COCO, the author found limited compositional performance but this wasn't really discussed any further.

**Questions:**

I didn't fully get why zero-shot is better than further training in MNIST. The authors say this is because of the difficulties of further training, but why?

The initialization of the code book is a very interesting observation. So how would one set this appropriately in practice to make sure it's successful? Should it always match the number of basic concepts?

---

> ### Author Response · Authors · 2024-11-19
> **Respnose to reviewers dKHK  (1)**
>
> We would like to thank the reviewers for recognizing our work as well-written and interesting, acknowledging its potential in zero-shot scenarios, and appreciating the comprehensive perspective on compositionality. Below, we provide detailed responses to all the questions raised by the reviewers. In light of these clarifications, we kindly request the reviewers to reconsider their scores favorably.
>
> > "Several methods have been proposed and evaluated during recent years by the emergent communication (EC) research community..." -> isn't meta-learning another one to mention here?
>
> While we recognize that meta-learning is a valuable approach for promoting, and potentially establishing, compositionality, we do not specifically consider it a solution to the channel discretization problem. We have added a section to the related work (Appendix H.2) discussing this approach.
>
> > "Notably, our method achieves extremely high performance, characterized by perfect accuracy and compositionality, on multiple datasets." -> just say it achieves perfect accuracy, no need to say extremely high
>
> Done.
>
> > "we assert that before effectively composing basic concepts into complex ones, agents must acquire the ability to decompose complex concepts into basic ones" -> must sounds quite strong here. Do you really show that there is no other way to do this? Or is your method one way to accomplish compositionality?
>
> We acknowledge and replaced ‘must’ with ‘should’
> To the best of our knowledge, no other method reported so far, achieved perfect compositionality and accuracy on similar datasets.
>
> > You're using "employed" a lot when really you could just say "use"
>
> Changed some.
>
> > I didn't get how the beta of the loss function (game and codebook-tradeoff) was set at first but later it said that the authors "experimented with various values of β1 and β2, ranging from 0.1 to 1.0 and found no significant difference in results." What do you make of this? Why does it not make a big difference? Is this because the loss is dominated by one part in any case?
>
> The loss is not dominated by any single component; rather, both components play a crucial role, and the optimization process seeks to minimize both. Our observations indicate that there are no universal hyperparameter values that consistently improve both accuracy and compositionality scores across all experiments. While specific settings might enhance results for certain experiments, we chose to fix these hyperparameters to demonstrate that achieving high accuracy and compositionality scores does not require fine-tuning them.
>
> > There is a lot of things going on in Figure 3. Why are there arrows in there? This seems to mostly just show exactly what is in the text already.
>
> We added the arrows to assist the reader in navigating the figure more easily.
>
> > I like the ablation studies although it could be a bit better explained why multi-target makes a difference and how the differences between the data sets in this ablation can be explained.
>
> We believe that we address this in Appendix D, where we discuss these aspects. We’re happy to provide additional details if anything remains unclear.
>
> > The paper discusses a multi-loss optimization, but the complexity and potential sensitivity of the CtD approach to hyperparameter settings (e.g., β1 and β2 values) could make it less accessible for broader use.
>
> For that, we fixed both β1 and β2 values to 1.0. See more details in previous answer.
>
> > For some datasets, such as COCO, the author found limited compositional performance but this wasn't really discussed any further.
>
> We further refer to this phenomenon in Appendix C.5. (lines 1329-1330), and in Appendix D.2 (lines 1421-1422). If there are remaining questions please let us know.
>
> > I didn't fully get why zero-shot is better than further training in MNIST. The authors say this is because of the difficulties of further training, but why?
>
> This question is challenging to address, and providing a definitive explanation will require further research. However, upon analyzing the detailed training results from five different runs, we observed that the optimization process failed to improve upon the initial starting point. We attribute this difficulty to the challenge of balancing the two competing losses on the newly introduced dataset during the composition step.
>
> > The initialization of the code book is a very interesting observation. So how would one set this appropriately in practice to make sure it's successful? Should it always match the number of basic concepts?
>
> We are pleased that you find the codebook initialization technique intriguing. As demonstrated, the reinitialization method described in Appendix E (lines 1430–1470) offers an effective approach to ensure proper codebook initialization and utilization. The codebook size does not need to precisely match the number of concepts. Additional ablation studies on this aspect are provided in Appendices E-1 and E-2.

---

> > ### Comment · Reviewer_dKHK · 2024-11-21
> > **Thank you.**
> >
> > Thank you very much for the clarifications. Given that these were mostly clarifications, so no larger changes to the paper, I think I will keep my initial score of a borderline accept.

---

> > > ### Author Response · Authors · 2024-11-26
> > > **Thanks to Reviewer dKHK for the Review**
> > >
> > > We sincerely appreciate the reviewers' thoughtful evaluation of our responses. In light of the reviewers' final comments, we kindly wish to inquire if a borderline acceptance accurately reflects the positive feedback provided.

---

### Author Response · Authors · 2024-11-19
**Revised version of the document with tracked changes for the reviewers.**

We have uploaded a revised version of the document that incorporates the reviewers' comments and suggestions. To facilitate easier tracking, the changes are highlighted in blue. The font will be reverted to its original style at the conclusion of the review period.

---

### Meta-Review · Area_Chair_o9AV · 2024-12-24

**Metareview:**

This paper introduces a method to have agents achieve compositional generalization by decompose and recompose known concepts. Compositional generalization is widely regarded as a key aspect of human intelligence to achieve great generalization. However, its mechanism and how it can be modeled remains a big challenge. The method is inspired by linguistic encoding and its use of discrete codebook is well contextualized and grounded. The experiments across various datasets justify and corroborate the method’s effectiveness. The metrics of the evaluation are well presented. Yep, first in order for the method to work it seems to have good prior knowledge (but it could be outside the scope of this paper’s resigner). In some parts of the writing, it is too dense (including figure 3) to fully grasp the flow. A few reference should be added (e.g., multi-generation-based approach). Overall, it is a solid paper with mostly good writing.

**Additional Comments On Reviewer Discussion:**

The main concern is about some details and references of this paper. The issues are mostly addressed in the discussion phase.

---

### Decision · Program_Chairs · 2025-01-22

Accept (Poster)